# BiTrajDiff: Bidirectional Trajectory Generation with Diffusion Models for Offline Reinforcement Learning

Yunpeng Qing [* 1]  Yixiao Chi [* 1]  Shuo Chen [* 1]  Shunyu Liu [2]  Kexuan Zhou [1]  Sixu Lin [3]  Litao Liu [4]  Changqing Zou [† 5 1]

## Abstract

Offline Reinforcement Learning (RL) relies on static datasets and often enforces conservative constraints to mitigate out-of-distribution errors, but this inevitably gives rise to learning dataset biases and limited behavioral generalization. Recent Data Augmentation (DA) methods leverage generative models to enrich offline data, yet they mainly operate within a single rollout paradigm and tend to preserve the original trajectory-level connectivity of the dataset. As a result, such methods often introduce local variations and fail to recover connections between distinct behavior patterns. In this paper, we propose Bidirectional Trajectory Diffusion (BiTrajDiff), a novel DA framework that explicitly addresses this limitation. BiTrajDiff decomposes trajectory synthesis into two independent diffusion processes that generate forward-future and backward-history segments conditioned on shared intermediate anchor states. By stitching the generated segments at these anchors, BiTrajDiff can synthesize trajectories that bridge disconnected behavior patterns and recover global trajectory-level connectivity absent from the original data. Extensive experiments demonstrate that BiTrajDiff consistently outperforms advanced DA methods across a range of offline RL backbones. Our code is available at https://github.com/Plankson/BiTrajDiff.

---

[*]Equal contribution ; [†]Corresponding author. [1]Zhejiang University [2]Nanyang Technological University [3]The Chinese University of Hong Kong (Shenzhen) [4]Rutgers University-New Brunswick [5]Zhejiang Lab. Correspondence to: Changqing Zou <changqing.zou@zju.edu.cn>.

*Proceedings of the 43rd International Conference on Machine Learning*, Seoul, South Korea. PMLR 306, 2026. Copyright 2026 by the author(s).

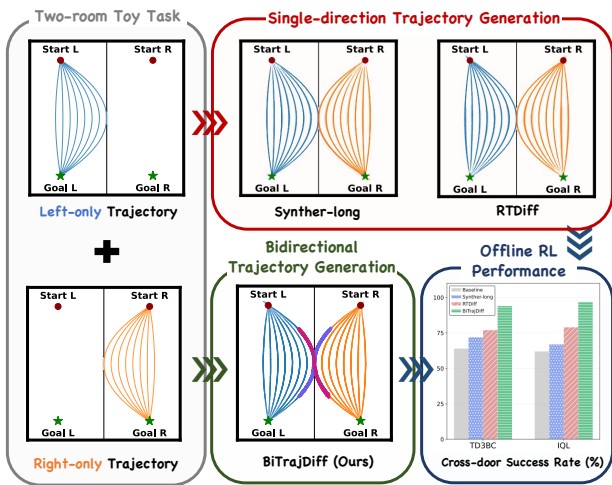

*Figure 1.* Two-room toy task with the offline dataset. The dataset contains **left-only** and **right-only** trajectories and lacks cross-door pattern; single-direction generation fits the in-room distribution but rarely bridges the doorway. BiTrajDiff generates **cross-door** trajectories and improves offline RL cross-door success.

## 1. Introduction

Offline Reinforcement Learning (Offline RL) (Fujimoto et al., 2019) seeks to derive effective decision policies solely from pre-collected datasets, obviating the necessity for online exploration due to the inherent risks and costs associated with real-time interaction (Levine et al., 2020). This paradigm has shown strong potential in real-world domains such as robotic control (Qing et al., 2024; 2022; Liu et al., 2023; Kong et al., 2024; Liu et al., 2024a) and power grid management (Chen et al., 2024; Xu et al., 2024). A primary challenge in offline RL is the out-of-distribution (OOD) problem: during training, the learned policy induces a state–action distribution that can deviate from the support of the logged data, causing extrapolation error and value overestimation (Levine et al., 2020). To mitigate this, many offline RL methods enforce conservatism by imposing explicit constraint terms during policy optimization (Qing et al., 2024; Chen et al., 2022) and penalizing the value estimation of OOD actions (Lyu et al., 2022; Xu et al., 2023). However, such conservatism also tightly induces learning to the

structural biases of the offline dataset, limiting behavioral diversity and generalization beyond the logged trajectories.

To alleviate the structural bias imposed by conservative offline learning, a natural solution, Data Augmentation (DA), enriches the offline dataset with synthesized transitions while remaining orthogonal to the underlying offline RL optimization procedure. Recent works (Lu et al., 2023; Li et al., 2024; Li & Zhang, 2024; Su et al., 2023) leverage generative models, particularly diffusion models, to synthesize high-fidelity trajectories for offline RL, yielding promising empirical improvements over existing offline RL algorithms. However, existing DA methods typically generate trajectories within a single rollout paradigm (forward-future (Lee et al., 2024) or backward-history (Yang & Wang, 2025)) conditioned on in-dataset states. Therefore, the synthesized data remain confined to the original trajectory distribution, rather than creating novel trajectories connecting previously disconnected states in the dataset. Consequently, the augmented data inevitably ignores the connectivity between various behavior patterns and fails to synthesize trajectories that explicitly bridge them. To exemplify the above issue, we conduct a didactic experiment using representative single-direction data augmentation methods, including Synther-long, a trajectory-level extension of Synther (Lu et al., 2023), and RTDiff (Yang & Wang, 2025), in a toy task shown in Figure 1. The offline dataset contains two distinct trajectory sets confined to each side of the wall and includes no cross-door trajectories, despite covering many states near the narrow doorway. The generation results show that the methods both successfully fit the distribution of the original data, capturing within-room variations, while rarely synthesizing trajectories that traverse the door to connect the two patterns. This indicates that single-process rollouts tend to preserve the original trajectory-level connectivity, resulting in limited performance improvement.

In this paper, we propose Bidirectional Trajectory Diffusion (BiTrajDiff), a novel DA framework that explicitly addresses the trajectory-level connectivity limitation of existing DA methods. Unlike prior approaches that rely on a single rollout paradigm, BiTrajDiff decomposes trajectory synthesis into two independent yet complementary diffusion generation processes: one modeling forward-future trajectories and the other modeling backward-history trajectories. Crucially, these two processes are conditioned on the same intermediate anchor states sampled from the dataset, enabling the independent exploration of plausible pasts and futures around these anchors. Therefore, BiTrajDiff can synthesize trajectories that bridge distinct behavior patterns and recover connections absent from the original dataset. As evidenced in Figure 1, despite the absence of any cross-door trajectories in the dataset, BiTrajDiff synthesizes such trajectories and achieves higher performance than the single-direction DA baselines. Technically, we implement

BiTrajDiff with a forward trajectory diffusion model conditioned on terminal states and a backward trajectory diffusion model conditioned on initial states, followed by a stitching procedure that concatenates the generated segments into complete trajectories. Meanwhile, we incorporate a lightweight quality-control step to discard low-confidence synthesized trajectories. As a result, BiTrajDiff expands the offline dataset not only through local variations but also by recovering global trajectory-level connections, leading to substantially improved performance across a wide range of offline RL algorithms.

**Our contributions** are summarized as follows:

- We identify a fundamental limitation of existing DA methods for enhancing offline RL algorithms: single rollout processes tend to preserve the original trajectory-level connectivity of the dataset, failing to explore the ways to connect distinct behavior patterns.

- We propose Bidirectional Trajectory Diffusion (BiTrajDiff), which decomposes trajectory synthesis into independent forward-future and backward-history diffusion processes, and then explicitly stitches the generated segments at shared intermediate anchor states, enabling the synthesis of trajectories that connect previously disconnected behavior patterns.

- We demonstrate the effectiveness of BiTrajDiff through extensive experiments on D4RL across multiple offline RL backbones, where it consistently outperforms various types of DA baselines, validating its performance gains and robustness.

## 2. Related Works

**Offline RL** encompasses four primary categories: policy constraint (Wang et al., 2022; Xu et al., 2023), value regularization (Kostrikov et al., 2021; Kumar et al., 2020), model-based (Yu et al., 2020; Chemingui et al., 2024), and return-conditioned supervised learning (Chen et al., 2021; Paster et al., 2022). Policy constraint methods restrict policies to the offline dataset, employing techniques such as explicit behavior cloning regularization (Qing et al., 2024; Chen et al., 2022) or implicit optimization towards in-sample optimal policies (Nair et al., 2020; Yue et al., 2022). Meanwhile, value regularization promotes conservative value functions to alleviate OOD overestimation via establishing precise lower bounds for Q-values to facilitate in-sample action selection (Lyu et al., 2022; Park et al., 2024). Model-based approaches construct task-oriented world models from offline data, including one-step dynamics (Yu et al., 2020; Chemingui et al., 2024) and multi-step sequence models (Hafner et al., 2020; 2023), which are subsequently used for offline reward penalization (Yu et al., 2021; Kidambi et al., 2020) and online decision planning to ensure in-distribution actions (Hansen et al., 2022; 2023).

Return-conditioned supervised learning trains trajectory generators conditioned on returns, with transformers (Chen et al., 2021; Schmied et al., 2024; Kong et al., 2025) and diffusion models (Janner et al., 2022; Ajay et al., 2022) as common backbones. Despite their effectiveness, these approaches are fundamentally limited by reliance on the dataset distribution.

**Data Augmentation for Offline RL** synthesizes additional interactions to enrich the dataset for subsequent RL training. Early work employs dynamics ensembles to generate transitions with uncertainty weighting (Zhang et al., 2023) or diffusion models for flexible one-step oversampling (Lu et al., 2023). More recent efforts shift toward trajectory-level generation. Some work focuses on on-policy rollouts, guided either by policy outputs (Jackson et al., 2024), return value (Lee et al., 2024), or interaction histories (He et al., 2023). Other approaches explore prioritized augmentation based on dynamics error or curiosity (Wang et al., 2024), as well as trajectory recomposition through reverse generation (Yang & Wang, 2025) or diffusion-based stitching (Li et al., 2024). Existing data augmentation methods for offline RL typically rely on a single generative process within a single forward-future or backward-history rollout paradigm, which preserves the original trajectory-level connectivity of the dataset. In contrast, BiTrajDiff synthesizes both forward and backward trajectories conditioned on shared intermediate states, allowing it to bridge distinct behavior modes among datasets and recover global connectivity, resulting in substantial improvements in offline RL performance.

## 3. Preliminary

**RL Paradigm** (Sutton et al., 1998) is formally defined as a Markov Decision Process (MDP), $\mathcal{M} = \langle \mathcal{S}, \mathcal{A}, P, r, \gamma, \rho_0 \rangle$, where $\mathcal{S}$ is the state space, $\mathcal{A}$ is the action space, $P : \mathcal{S} \times \mathcal{A} \times \mathcal{S} \to [0, 1]$ represents the environment dynamics, $r : \mathcal{S} \times \mathcal{A} \to \mathbb{R}$ is the reward function, $\gamma \in [0, 1)$ is the discount factor, and $\rho_0$ denotes the initial state distribution. At each timestep $t$, an agent executes an action $a_t$ according to its policy $\pi(\cdot|s_t)$ in the current state $s_t$, transitioning to the subsequent state $s_{t+1}$ with reward $r_t$ based on $P$. The objective of the agent is to find an optimal policy $\pi^*$ that maximizes the expected discounted return: $\pi^* = \arg\max_\pi \mathcal{J}(\pi) = \mathbb{E}[\sum_{t=0}^\infty \gamma^t r_t]$. In contrast to online RL, offline RL (Levine et al., 2020) relies solely on a static trajectories dataset $\mathcal{D} = \{\tau^i\}_{i=1}^N$, where $N$ is the number of trajectories and each trajectory $\tau^i = \{(s_t^i, a_t^i, r_t^i)\}_{t=0}^{T-1}$ comprises a sequence of state-action-reward tuples.

**Diffusion Models** (Ho et al., 2020; Song et al., 2021) are generative frameworks that learn data distributions through a forward diffusion process and a reverse denoising process. The forward process gradually corrupts data $\mathbf{x}_0$ via a pre-

defined noise schedule $\{\beta_k\}_{k=1}^K$, where each transition is defined as $q(\mathbf{x}_k|\mathbf{x}_{k-1}) = \mathcal{N}(\mathbf{x}_k; \sqrt{1-\beta_k}\mathbf{x}_{k-1}, \beta_k\mathbf{I})$. As $K \to \infty$, the terminal distribution $q(\mathbf{x}_K)$ converges to an isotropic Gaussian $\mathcal{N}(\mathbf{0}, \mathbf{I})$. The reverse process, parameterized by $\theta$, aims to iteratively reconstruct the original data through learnable Gaussian transitions: $p_\theta(\mathbf{x}_{k-1}|\mathbf{x}_k) = \mathcal{N}(\mathbf{x}_{k-1}; \mu_\theta(\mathbf{x}_k, k), \Sigma_\theta(\mathbf{x}_k, k))$. By maximizing the empirical lower bound (Sohn et al., 2015) on the log-likelihood of the sampled data, the diffusion model can be trained with a simplified surrogate loss (Sohn et al., 2015):

$$\mathcal{L}_{\text{denoise}}(\theta) = \mathbb{E}_{\mathbf{x}_0 \sim q, k \sim \mathcal{U}\{1,K\}, \epsilon \sim \mathcal{N}(\mathbf{0},\mathbf{I})} \left[ \|\epsilon_\theta(\mathbf{x}_k, k) - \epsilon\|_2^2 \right], \quad (1)$$

where $\mathcal{U}$ denotes the discrete uniform distribution, and $\epsilon_\theta$ is the deep model parameterized with $\theta$ to predict the noise. Both the mean $\mu_\theta(\mathbf{x}_k, k)$ and covariance $\Sigma_\theta(\mathbf{x}_k, k)$ of the reverse process are analytically derivable from $\epsilon_\theta$. The sampling is then performed by initializing with $\mathbf{x}_K \sim \mathcal{N}(\mathbf{0}, \mathbf{I})$ and iteratively applying the learned reverse transitions.

## 4. Methodology

This section introduces the BiTrajDiff framework for offline RL, which comprises two core components: *bidirectional diffusion training* and *bidirectional trajectory generation*. During the *bidirectional diffusion training* phase, two distinct diffusion models are employed to model the distributions of forward-future and backward-history state sequences within trajectory data. Conditioned on an intermediate state and cumulative reward signals, these models generate state sequences in corresponding temporal directions. Then, in the subsequent *bidirectional trajectory generation* phase, the generated forward and backward trajectories are reconciled through filling and filtering operations to produce novel global trajectories that connect previously unreachable states in the original dataset. The framework of our method is presented in Figure 2.

### 4.1. Bidirectional Diffusion Training

Building upon recent advances in diffusion-based sequence generation (Fathi et al., 2025; Gong et al., 2022), we propose a bidirectional trajectory generation framework comprising independent forward and backward trajectory diffusion models. These models are separately trained on offline datasets to learn the trajectory distribution of the behavior policy, facilitating bidirectional state trajectory generation from shared intermediate "anchor" states.

Formally, for each diffusion model, the state trajectory is generated simultaneously per time step $t$ over the planning horizon $H$: $\mathbf{x}_k(\tau) = \{s_t, s_{t+1}, \cdots, s_{t+H-1}\}_k$, where $k$ denotes the denoise timestep and $t$ represents the MDP timestep. In this way, the conditional generative problem

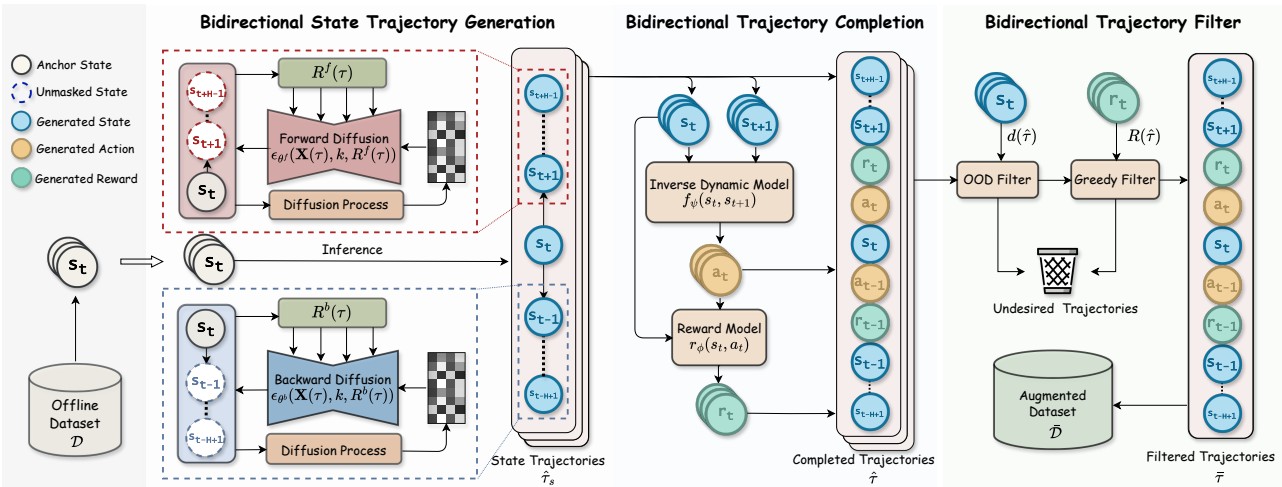

*Figure 2.* An illustrative diagram of our *Bi*directional *Traj*ectory *Diff*usion (BiTrajDiff) method.

via the diffusion model can be formulated as:

$$\min_\theta \mathcal{L}(\theta) = \mathbb{E}_{\tau \sim \mathcal{D}}[-\log p_\theta(\mathbf{x}_0(\tau)|\mathbf{y}(\tau))], \quad (2)$$

where $\mathbf{x}_0(\tau)$ denotes the final generated subtrajectory of diffusion, $\mathbf{y}(\tau)$ represents the condition for generation, and $\theta$ represents the learnable parameters of the diffusion model. While various condition metrics are available, such as temporal difference error, Q-function (Wang et al., 2024), our framework utilizes the cumulative return $R$ as conditional input $\mathbf{y}(\tau)$. This approach, consistent with prior works (Ajay et al., 2022; Ding et al., 2024), enables the generation of trajectories with desired returns starting from the specific states. However, to enable bidirectional trajectory generation, the formations of the backward and forward diffusion condition inputs are quite distinct.

Specifically, the forward diffusion model generates finite-horizon state trajectories $\mathbf{x}_0^f(\tau) = \{s_t, s_{t+1}, \cdots, s_{t+H-1}\}$ **started at initial state $s_t$ with target return** $R^f(\tau) = \sum_{i=t}^\infty \gamma^{i-t} r_i$, thus generating physically consistent future trajectories within the offline distribution. Conversely, the backward diffusion model reconstructs historical trajectories $\mathbf{x}_0^b(\tau) = \{s_{t-H+1}, \cdots, s_t\}$ that **ends on terminal state $s_t$ and accrued return** $R^b(\tau) = \sum_{i=0}^t \gamma^{t-i} r_i$. For brevity and clarity, we utilize superscript-free notation (e.g., $\mathbf{x}_k$, $\mathbf{y}$, $R(\tau)$) in the following sections to denote computational processes common to both directional models.

Based on the two distinct conditions, we can model the conditional trajectory generation processes by the diffusion sampling composed of the forward noising process $q(\mathbf{x}_{k+1}(\tau)|\mathbf{x}_k(\tau))$ and the reverse denoising process $p_\theta(\mathbf{x}_{k-1}(\tau)|\mathbf{x}_k(\tau), \mathbf{y}(\tau))$ constructed with the diffusion denoise model $\epsilon_\theta$. Meanwhile, the forward and backward diffusion models $\epsilon_\theta$ in our BiTrajDiff framework are parameterized with $\theta^f$ and $\theta^b$, respectively. Thus, given the

condition $\mathbf{y}(\tau)$, they are able to generate the denoised trajectory $\mathbf{x}_0(\tau)$. This process begins with sampling random noise $\mathbf{x}_K \sim \mathcal{N}(\mathbf{0}, \mathbf{I})$, followed by iterative denoising steps where $\mathbf{x}_{k-1} \sim p_\theta$, for $k = \{K, \cdots, 1\}$.

To realize conditional diffusion model training and inference, we employ the Classifier-Free Guidance (CFG) (Ho & Salimans, 2022), which uses a single denoise model to handle both conditional and unconditional generation. Formally, the perturbed noise after CFG can be formed as:

$$\hat{\epsilon} = \omega * \epsilon_\theta(\mathbf{x}_k(\tau), k, \mathbf{y}(\tau)) + (1 - \omega) * \epsilon_\theta(\mathbf{x}_k(\tau), k, \emptyset), \quad (3)$$

where the scalar $\omega \in [0, 1]$ denotes the guidance weight of CFG. Setting $\omega$ to 0 degrades the generation to unconditional generation. Conversely, an increased value of $\omega$ promotes a stronger adherence of the generated samples to the provided condition $\mathbf{y}(\tau)$. During training, the diffusion denoisers are trained with an objective that enables them to predict noise conditioned on $\mathbf{y}(\tau)$ and also unconditionally. This is typically achieved by randomly dropping the condition $\mathbf{y}(\tau)$ during training with probability $p$. Specifically, we have the following loss function:

$$\mathcal{L}_{\text{denoise}}(\theta) = \mathbb{E}_{\tau \sim \mathcal{D}, k \sim \mathcal{U}\{1, K\}, \epsilon \sim \mathcal{N}(\mathbf{0}, \mathbf{I}), \beta \sim \text{Bern}(p)}$$
$$\left[ \| \epsilon - \epsilon_\theta(\mathbf{x}_k(\tau), k, (1 - \beta)\mathbf{y}(\tau) + \beta\emptyset) \|^2 \right], \quad (4)$$

where $\text{Bern}(p)$ represents the Bernoulli distribution. In our implementation, we fix the state condition $s_t$ in $\mathbf{x}_k(\tau)$ during the adding-noise training as well as the denoising inference stage for simplicity and efficiency following (Ajay et al., 2022; Yang & Wang, 2025), while the diffusion denoise models $\epsilon_\theta$ only take the cumulative return $R(\tau)$ components of $\mathbf{y}(\tau)$ as condition inputs.

## 4.2. Bidirectional Trajectory Generation

Based on the pretrained bidirectional diffusion model, we introduce a novel DA pipeline through three key stages: bidirectional state trajectory generation, bidirectional trajectory completion, and bidirectional trajectory filtering. The pipeline synthesizes global trajectories, thereby establishing connections between states not accessible in the datasets. This process extends behavioral coverage while upholding physical consistency, ultimately producing effective augmented data for offline RL algorithms training.

### 4.2.1. BIDIRECTIONAL TRAJECTORY GENERATION

We utilize the pretrained forward and backward trajectory diffusion model in section 4.1 to generate bidirectional state trajectories. Prior methods (Lu et al., 2023; Li & Zhang, 2024) model only the conditional distribution of trajectories originating from a given state, neglecting the historical pathways leading to those states. As a result, these data augmentation techniques yield only limited expansion in behavior diversity. To address this limitation without introducing additional noise, our proposed method, BiTrajDiff, generates both forward and backward trajectories and integrates them by stitching at a shared intermediate state $s_t$, which serves as a consistent anchor.

Specifically, we sample the candidate state $s$ from the offline dataset $\mathcal{D}$ as the condition state $s_t$. By combining the state $s_t$ with the manually selected return signal $R(\tau)$ into the $\mathbf{y}(\tau)$, we can leverage the pre-trained bidirectional diffusion model $\theta$ to generate two distinct trajectories: a forward-future trajectory $\mathbf{x}_0^f(\tau)$ originates from $s_t$, and a backward-history trajectory $\mathbf{x}_0^b(\tau)$ concludes at $s_t$. Since $s_t$ is both the start state of $\mathbf{x}_0^f(\tau)$ and the end state of $\mathbf{x}_0^b(\tau)$, we can directly stitch them to construct the bidirectional state trajectory $\hat{\tau}_s(s_t)$. Formally, let $\mathbf{x}_0(\tau)[i:j]$ denote the observation slice from time $i$ to time $j$, the stitched bidirectional state trajectory $\hat{\tau}_s(s_t)$ can be formulated as:

$$
\begin{aligned}
\hat{\tau}_s(s_t) &= \left\{ \mathbf{x}_0^b(\tau)[0:H-1], s_t, \mathbf{x}_0^f(\tau)[1:H] \right\} \\
&= \left\{ \hat{s}_{t-H+1}, \cdots, \hat{s}_{t-1}, s_t, \hat{s}_{t+1}, \cdots, \hat{s}_{t+H-1} \right\},
\end{aligned}
\tag{5}
$$

where $H$ is the horizon, and $\hat{s}_i$ is the diffusion generated states at MDP timestep $i$. By conditioning on and concatenating at the intermediate anchor state $s_t$, we construct a state trajectory, $\hat{\tau}_s(s_t)$, that connects $\hat{s}_{t-H+1}$ and $\hat{s}_{t+H-1}$. This process enables the connection of states that may be disconnected in the original dataset, thereby substantially enriching the diversity of observed behavioral patterns. Appendix K provides a formal interpretation of the trajectory-level connectivity gap between our bidirectional paradigm and existing single-direction paradigms, and further explains why bidirectional stitching is able to recover couplings that cannot be captured by single-direction trajectory rollouts.

### 4.2.2. BIDIRECTIONAL TRAJECTORY COMPLETION

In this completion process, the generated bidirectional state trajectories $\hat{\tau}_s(s_t)$ are further completed by adding the action and reward signal. We introduce the Inverse Dynamics Model (IDM): $f_\psi(s_t, s_{t+1}) : \mathcal{S} \times \mathcal{S} \to \mathcal{A}$ and the Reward Model (RM): $r_\phi(s_t, a_t) : \mathcal{S} \times \mathcal{A} \to \mathbb{R}$. The supervised training objective of the two models is:

$$
\begin{aligned}
\mathcal{L}(\psi, \phi) = \mathbb{E}_{(s_t, a_t, r_t, s_{t+1}) \sim \mathcal{D}} \Big[ & \| f_\psi(s_t, s_{t+1}) - a_t \|^2 \\
& + \| r_\phi(s_t, a_t) - r_t \|^2 \Big].
\end{aligned}
\tag{6}
$$

Thus, by feeding the generated state trajectory $\hat{\tau}_s(s_t)$ to the trained IDM and RM sequentially, we obtain the completed bidirectional trajectory : $\hat{\tau}(s_t) = \{(\hat{s}_i, \hat{a}_i, \hat{r}_i, \hat{s}_{i+1})\}_{i=t-H+1}^{t+H-2}$, where $\hat{a}_i = f_\psi(\hat{s}_i, \hat{s}_{i+1})$ and $\hat{r}_i = r_\phi(\hat{s}_i, \hat{a}_i)$. Therefore, we can obtain a generated dataset $\hat{\mathcal{D}} = \{\hat{\tau}(s_{t_i})\}_{i=1}^N$ with $N$ generated trajectories.

### 4.2.3. BIDIRECTIONAL TRAJECTORY FILTER

To ensure dataset quality, we employ a two-stage filtering mechanism that removes both out-of-distribution and suboptimal trajectories. This selective process preserves the reliability of the augmented data without sacrificing diversity. In BiTrajDiff, the trajectory filter comprises an OOD trajectory filter and a greedy trajectory filter.

**OOD Trajectory Filter.** We model the OOD trajectory filter as a novelty detection model (Pimentel et al., 2014), which addresses the identification of data instances that exhibit a significant deviation from the offline training dataset. And we utilize the Isolation Forest model (Liu et al., 2008) for OOD transition detection, which assigns an anomaly score to each data instance, reflecting its susceptibility to isolation with shorter path lengths in randomly partitioned trees. Specifically, we construct the isolation forest on the original dataset $\mathcal{D}$, which is able to give each observation $\hat{s}_i$ an anomaly score $d(\hat{s}_i)$. We defined the OOD score of a generated trajectory $d(\hat{\tau}) = \sum_{i=t-H+1}^{t+H-2} d(\hat{s}_i)$. Then we sort $\hat{\mathcal{D}}$ by $d(\hat{\tau})$ and retain the top-$C_{\text{ood}}$ trajectories with the smallest values, where $C_{\text{ood}}$ is a hyperparameter.

**Greedy Trajectory Filter.** The greedy trajectory filter selects the generated trajectories with the highest cumulative reward. Specifically, we sort the $C_{\text{ood}}$ trajectories retained from OOD trajectory filter by the sum reward $R(\hat{\tau}) = \sum_{i=t-H+1}^{t+H-2} \hat{r}_i$. The top-$C_{\text{greedy}}$ trajectories with the highest sum rewards are picked as the final generated trajectories dataset $\tilde{\mathcal{D}}$ for offline RL training.

In summary, BiTrajDiff operates in two key stages. Initially, it trains a bidirectional diffusion model to capture the distributions of forward-future and backward-history trajectories.

*Table 1.* Performance of our BiTrajDiff and baselines on the locomotion tasks. ± corresponds to the standard deviation of the performance on 5 random seeds. The best and the second-best results of each setting are marked as **bold** and underline, respectively. Detailed reports about CQL and DT are reported in Appendix C.

| Source | Task | IQL (Kostrikov et al., 2021) | | | | | TD3BC (Fujimoto & Gu, 2021) | | | | |
|---|---|---|---|---|---|---|---|---|---|---|---|
| | | Base | RTDiff | Synther | DiffStitch | Ours | Base | RTDiff | Synther | DiffStitch | Ours |
| medium | halfcheetah | 48.2±0.2 | 48.9±0.2 | **49.2±0.2** | 48.6±0.2 | 48.6±0.2 | 48.4±0.2 | 49.4±0.2 | 49.7±0.4 | 50.1±0.6 | **50.3±0.4** |
| | hopper | 67.0±1.9 | 62.0±2.4 | 61.1±2.7 | 65.5±4.7 | **81.1±5.0** | 60.4±3.7 | 62.6±5.1 | 61.2±6.1 | 67.9±3.7 | **79.0±3.5** |
| | walker2d | 77.6±5.3 | 82.7±4.0 | 85.8±0.7 | 79.8±1.8 | **86.7±1.7** | 82.0±2.7 | 85.4±1.1 | 83.4±2.0 | 85.5±0.7 | **86.7±1.0** |
| medium expert | halfcheetah | 90.1±4.4 | 92.0±3.1 | 86.6±6.2 | 90.3±8.0 | **95.3±0.1** | 92.3±5.1 | 93.9±2.1 | 95.5±0.8 | 95.9±0.7 | **96.3±0.4** |
| | hopper | 105.4±6.3 | 110.5±0.5 | **111.2±0.7** | 108.6±1.2 | 110.9±0.6 | 94.2±11.1 | 106.8±7.0 | 106.2±6.7 | 97.5±9.9 | **109.0±5.1** |
| | walker2d | 112.4±0.4 | 111.9±0.5 | 112.1±0.6 | 111.2±0.3 | **112.5±0.5** | 109.3±0.2 | 109.6±0.4 | 109.4±0.0 | 109.6±0.5 | **110.2±0.3** |
| medium replay | halfcheetah | 43.5±0.2 | 43.7±0.8 | **46.2±0.3** | 43.6±0.1 | 44.1±0.2 | 44.1±0.1 | 44.0±0.3 | 44.5±0.6 | **45.1±0.5** | 45.0±0.6 |
| | hopper | 94.7±4.0 | 97.7±2.7 | 100.6±0.4 | 96.4±5.6 | **102.8±0.6** | 64.4±8.9 | 72.1±18.9 | 76.4±3.0 | 71.5±21.0 | **85.0±10.3** |
| | walker2d | 73.3±7.5 | 74.0±5.2 | 86.7±1.1 | 74.9±8.2 | **87.6±2.2** | 80.9±6.0 | 83.5±7.4 | 81.1±5.9 | 80.5±5.2 | **89.9±1.4** |
| **Average** | | 79.1 | 80.4 | 82.2 | 79.9 | **85.5** | 75.1 | 78.6 | 78.6 | 78.2 | **83.5** |

*Table 2.* Performance of our BiTrajDiff and baselines on the navigation and manipulation tasks.

| Task | IQL (Kostrikov et al., 2021) | | | | | TD3BC (Fujimoto & Gu, 2021) | | | | |
|---|---|---|---|---|---|---|---|---|---|---|
| | Base | RTDiff | Synther | DiffStitch | Ours | Base | RTDiff | Synther | DiffStitch | Ours |
| maze2d-umaze | 56.0±6.1 | 53.1±2.2 | 51.9±4.8 | 52.1±5.9 | **62.3±2.7** | 38.1±10.2 | 43.1±6.4 | 40.3±10.0 | 42.5±8.3 | **45.3±3.1** |
| maze2d-medium | 41.2±2.9 | 50.8±9.6 | 66.9±16.9 | 46.4±10.2 | **72.2±14.8** | 30.2±16.3 | 36.3±9.9 | 34.8±14.2 | 34.7±3.0 | **45.7±7.6** |
| maze2d-large | 67.9±4.2 | 74.3±5.2 | 70.9±2.5 | 64.2±6.1 | 71.3±2.3 | 95.3±22.7 | 102.8±8.9 | 100.6±34.3 | 106.1±11.4 | **129.3±26.0** |
| antmaze-umaze-diverse | 53.0±4.7 | 61.6±5.7 | 55.4±14.2 | 55.4±5.4 | **63.4±7.9** | 43.2±7.4 | 45.8±4.4 | 45.0±10.3 | **49.4±4.8** | 47.0±2.1 |
| antmaze-medium-diverse | 71.2±5.0 | 73.2±8.8 | 81.2±6.4 | 69.2±17.2 | **86.2±5.8** | 0.0±0.0 | 0.0±0.0 | **6.4±5.9** | 0.0±0.0 | 4.8±1.7 |
| antmaze-large-diverse | 18.0±8.4 | 53.2±9.5 | 57.8±6.4 | 12.8±15.2 | **63.0±9.3** | 0.0±0.0 | 0.0±0.0 | 0.4±0.5 | 0.0±0.0 | **1.2±0.9** |
| **Maze Average** | 51.2 | 61.0 | 64.0 | 50.0 | **69.7** | 34.5 | 38.0 | 37.9 | 38.8 | **45.6** |
| kitchen-complete | 43.3±16.0 | 51.9±7.4 | 42.8±16.4 | 38.3±8.9 | **59.1±12.3** | 0.1±0.1 | 0.8±0.5 | 0.1±0.2 | 0.5±0.7 | **1.5±1.0** |
| kitchen-partial | 68.0±5.1 | 69.8±2.9 | **70.3±2.8** | 62.9±9.4 | 69.7±1.8 | 11.8±9.7 | 8.9±7.0 | 10.9±1.8 | **13.0±9.3** | 13.0±5.9 |
| kitchen-mixed | 60.7±2.5 | **64.4±6.1** | 62.6±2.7 | 63.4±5.2 | 62.9±4.9 | 8.7±5.1 | 7.7±0.7 | 10.7±5.0 | 6.0±4.7 | **30.9±12.6** |
| **Kitchen Average** | 57.3 | 62.0 | 58.6 | 54.9 | **63.9** | 6.9 | 5.8 | 7.2 | 6.5 | **15.1** |

Subsequently, BiTrajDiff generates bidirectional trajectories conditioned on anchor states sampled from $\mathcal{D}$. These generated sequences undergo further completion and filtering to create a high-fidelity augmented dataset $\tilde{\mathcal{D}}$, which is then mixed with $\mathcal{D}$ to train downstream offline RL algorithms. We adopt CleanDiffuser (Dong et al., 2024) as the backbone for BiTrajDiff, owing to its generality and robustness. The pseudocode of BiTrajDiff is presented in Appendix A.

# 5. Experiments

To demonstrate the effectiveness of the proposed BiTrajDiff, we conduct experiments on the D4RL benchmark (Fu et al., 2020). Our evaluation seeks to answer the following questions: (1) Does BiTrajDiff outperform existing DA methods across various offline RL algorithms? (Section 5.2 and Appendix C- D). (2) How do different components of BiTrajDiff affect the offline RL performance? (Section 5.3 and Appendix E-G) (3) Can BiTrajDiff find the potentially valuable state leading to high return more effectively than other DA baselines? (Section 5.4 and Appendix H) (4) Can

BiTrajDiff generate more diverse trajectories compared to single-direction DA approaches? (Section 5.5)

## 5.1. Experiment Settings

We evaluate our approach on the D4RL benchmark (Fu et al., 2020). We compare BiTrajDiff with three classical DA methods: Synther (Lu et al., 2023), DiffStitch (Li et al., 2024) and RTDiff (Yang & Wang, 2025). We select four representative offline RL algorithms as baselines: IQL (Kostrikov et al., 2021), TD3BC (Fujimoto & Gu, 2021), CQL (Kumar et al., 2020), and DT (Chen et al., 2021). These methods have been widely adopted as standard baselines in offline RL due to their stable performance. The hyperparameter setting can be viewed in Appendix B.2.

## 5.2. Main Results

**Results for Locomotion Tasks.** The experimental results on D4RL locomotion tasks are summarized in Table 1. Our proposed BiTrajDiff demonstrates consistent superiority in

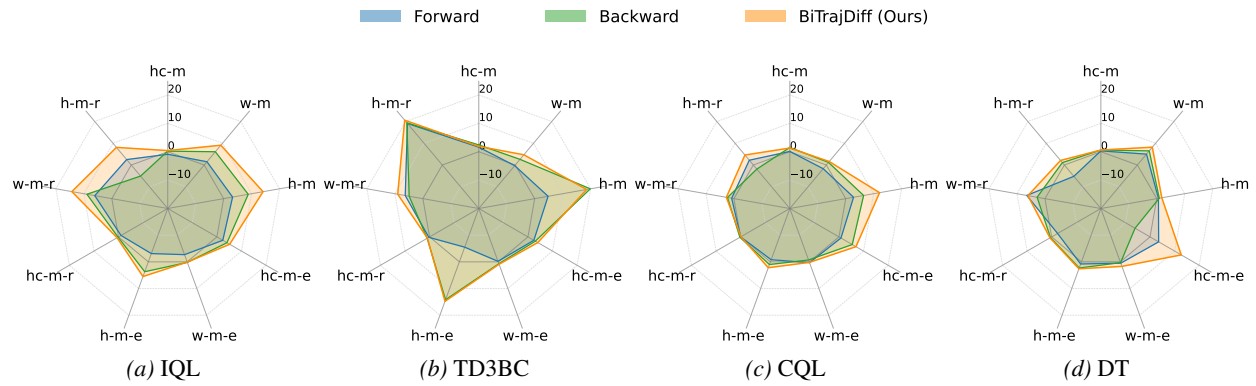

*Figure 3.* Performance improvement comparison of offline RL algorithms augmented with single- and bi-directional diffusion trajectories. The task abbreviations are listed in Table S1.

various task settings. Meanwhile, results on CQL and DT, presented in Appendix C, exhibit a similar phenomenon. These consistent gains across the offline RL algorithms underscore the effectiveness of BiTrajDiff.

**Results for Maze and Franka Kitchen Tasks.** Table 2 reports the experimental results on the D4RL Maze and Franka Kitchen tasks. BiTrajDiff achieves the highest average return under both the IQL and TD3BC algorithms. Since the Maze and Franka Kitchen tasks are sparse-reward settings, which are typically detrimental to RL training, the availability of high-quality data is particularly critical. BiTrajDiff tackles this challenge by enhancing data diversity quality, yielding superior and more stable performance.

### 5.3. Ablation Analysis

**Ablation Study of the Direction of Diffusion Generation.** To confirm the effectiveness of our bidirectional generation paradigm, we compare the performance improvement with the augmented diffusion trajectories generated in a single direction and in a bidirectional way, as shown in Figure 3. While the forward method yields moderate improvements, it is consistently outperformed because it neglects the historical transitions leading to critical, high-reward states. Conversely, the backward method achieves substantial gains by exploring trajectories toward a predefined initial state, thus mitigating the overestimation risk associated with unknown states (Yang & Wang, 2025). Finally, demonstrating consistently superior and robust performance, our bi-directional framework, BiTrajDiff, empirically validates that by simultaneously synthesizing forward-future and backward-history trajectories, BiTrajDiff constructs novel, high-quality paths absent from the original dataset.

**Ablation Study of Trajectory Filters.** We compare the IQL and TD3BC training process augmented by BiTrajDiff datasets with different trajectory filter strategies. The learning curves are presented in Figure 4. Without any filters, the training process is highly volatile, as unfiltered, potentially

OOD trajectories introduce suboptimal or even illegal states and cause erratic policy updates. In contrast, applying the OOD filter substantially stabilizes the training process by constraining synthetic data to a plausible behavioral space, yet this configuration consistently underperforms the full Bi-TrajDiff model. This is because the greedy filter selectively retains trajectories that yield higher returns, thereby actively guiding the policy toward good behaviors and facilitating the discovery of better behavior patterns.

**Ablation Study of Augmented Data Ratio.** We investigate the balance between original and augmented data by varying the ratio $\sigma$ of augmented to original data, with the results presented in Figure 5. When $\sigma$ is too low, performance improvements remain limited across the tested algorithms, illustrating that a limited amount of augmented data is not enough to provide offline RL algorithms with new behavior patterns. Conversely, increasing the ratio to a range of $30\% \sim 50\%$ leads to a substantial performance boost and stability. However, when the ratio is increased to $100\%$, the performance becomes highly unstable, which suggests that the excessive noise introduced by an overabundance of synthetic data hinders offline RL algorithms from learning effectively. Finally, we select $\sigma = 30\%$ as the fixed hyperparameter due to its stable performance improvement.

### 5.4. Effectiveness under $n$-step TD Estimator

We employ the $n$-step TD estimator to update the critic networks in IQL and TD3BC under the original dataset and the DA-augmented data, and further compare performance across different values of $n$ as shown in Figure 6. The $n$-step TD estimator (De Asis et al., 2018) enables the discovery of high-return solutions by looking ahead from more distant states, making performance improvement a natural indicator of the quality of DA-generated datasets. As shown in Figure 6, all three DA methods consistently outperform the base method, suggesting that the generated trajectories effectively uncover more rewarding states. How-

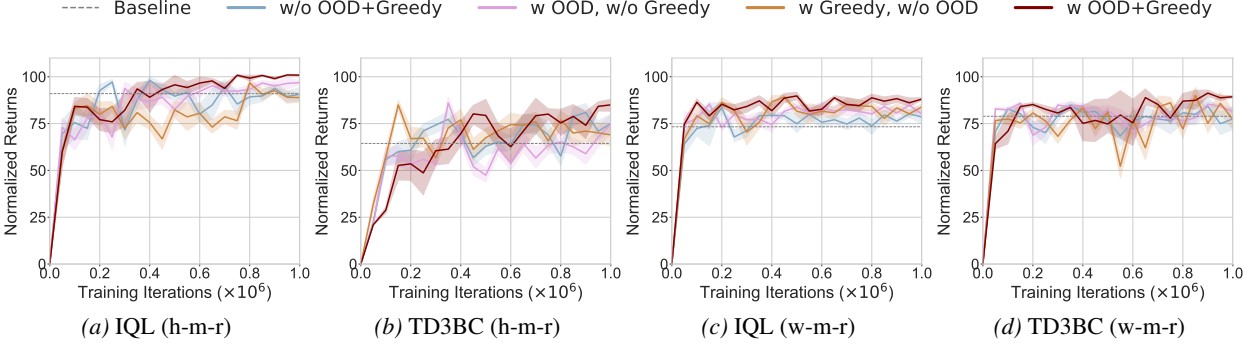

*Figure 4.* Learning curves of BiTrajDiff with different trajectory filter strategies. Detailed results are reported in Appendix F.

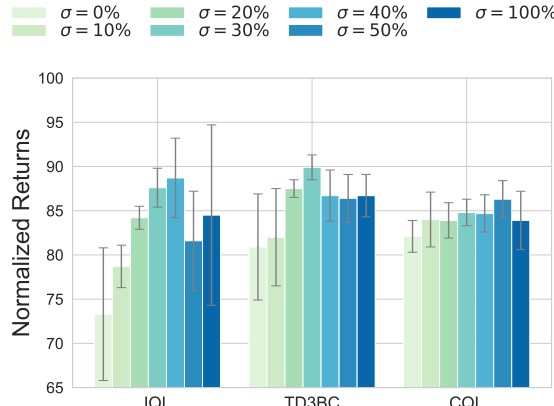

*Figure 5.* Compare the returns of BiTrajDiff with different augmented data ratios $\sigma$ in the walker2d-medium-replay task. Detailed results are reported in Appendix G.

ever, as $n$ increases, the performance inevitably declines due to the growing variance in value estimation. Meanwhile, for DA methods, this effect is compounded by the accumulated mismatching errors introduced by the inverse dynamics model, which negatively impact TD estimation. In particular, our method consistently achieves the highest performance across all tested settings and maintains its advantage over RTDiff and Diffstitch as the $n$-step horizon increases. These results verify the effectiveness of BiTrajDiff for offline RL algorithms under $n$-step TD estimator. By stitching forward-future and backward-history trajectories, BiTrajDiff obtains a superior capability to synthesize high-quality trajectories compared to existing DA methods.

### 5.5. Visualization

To evaluate the accuracy and diversity robustness of the BiTrajDiff framework, we compare its generated trajectories with those produced by single-direction forward and backward diffusion models. Following Lu et al. (2023), we employ two quantitative metrics to assess a generated trajectory $\hat{\tau}$: (1) Dynamic Error: $\mathcal{E}_{\text{Dyn}}(\hat{\tau}) = \sum_t \|\hat{s}_{t+1} - s_{t+1}\|_2^2$, which measures trajectory accuracy by summing

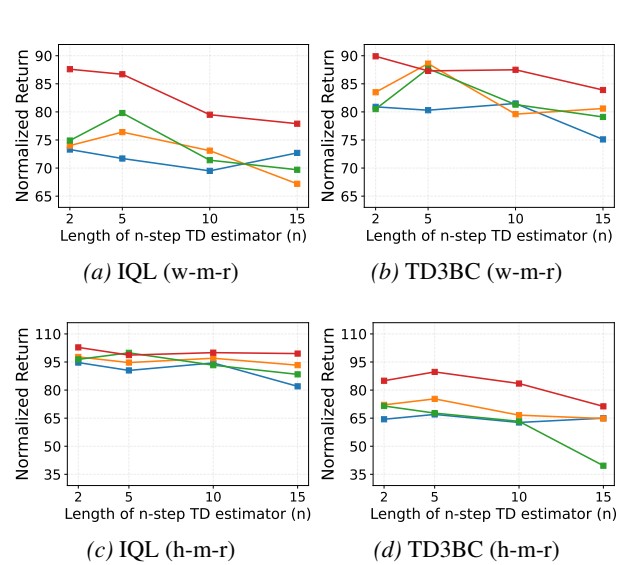

*Figure 6.* Comparisons of different methods under varying $n$-step TD estimators. Detailed results are reported in Appendix H.

the mean squared error between the predicted next states $\hat{s}_{t+1}$ and the ground truth $s_{t+1}$; (2) L2 Distance: $\mathcal{E}_{\text{L2D}}(\hat{\tau}) = \min_{\tau \in \mathcal{D}} \sum_t \|\hat{s}_t - s_t\|_2$, defined as the minimal L2 distance between the generated trajectory and any trajectory of equal length in $\mathcal{D}$, capturing trajectory diversity. As shown in Figure 7, while forward and backward diffusion models achieve low $\mathcal{E}_{\text{Dyn}}$, they exhibit limited $\mathcal{E}_{\text{L2D}}$, indicating that the generated trajectories closely resemble those in the original dataset and thus offer limited performance gains. In contrast, BiTrajDiff demonstrate significantly broader marginal distributions in $\mathcal{E}_{\text{L2D}}$ while maintaining competitive $\mathcal{E}_{\text{Dyn}}$, reflecting enhanced behavioral diversity without sacrificing dynamic consistency.

## 6. Conclusion

In this paper, we present BiTrajDiff, a novel diffusion-based framework that enhances offline RL by enabling bidirec-

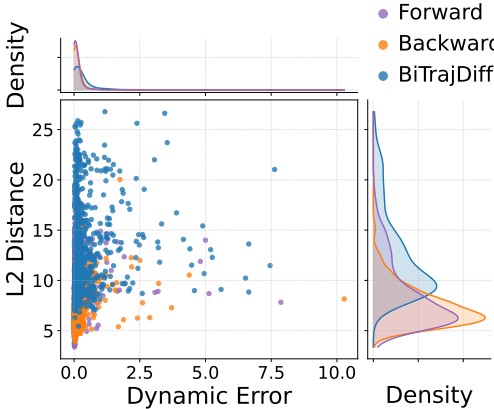

*Figure 7.* BiTrajDiff vs forward/backward diffusion: dynamic error and L2 distance: Comparisons of dynamic error and L2 distance.

tional trajectory generation from shared intermediate anchor states. Unlike conventional single rollout paradigms, which model forward or backward trajectories from observed states, BiTrajDiff generates both forward-future and backward-history trajectories, conditioned on shared intermediate anchor states, effectively bridging previously disconnected behavior patterns in the state space. This leads to significantly more diverse and effective trajectories, improving offline RL performance beyond the capability of other DA frameworks. Extensive experiments on the D4RL benchmark also demonstrate that BiTrajDiff significantly outperforms existing data augmentation baselines across diverse offline RL algorithms. Future work will explore bidirectional generation for offline RL under multi-task settings with broader behavioral patterns.

## Impact Statement

This paper presents work whose goal is to advance the field of offline reinforcement learning. There are many potential societal consequences of our work, none of which we feel must be specifically highlighted here.

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

## A. Pseudocode

The pseudocode of our BiTrajDiff method is provided in Algorithm 1.

---

**Algorithm 1** Bidirectional Trajectory Diffusion (BiTrajDiff) data augmentation pipeline.

---

**Input:** dataset $\mathcal{D}$, ratio $\sigma$, horizon $H$, target return-to-go $R^f$ and $R^b$, filter number $C_{\text{ood}}$ and $C_{\text{greedy}}$.
**Initialize:** Forward and backward diffusion denoise network $\epsilon_{\theta f}$ and $\epsilon_{\theta b}$, inverse dynamics model $f_\psi$, reward network $r_\phi$, and the augmented dataset $\mathcal{D}_{\text{da}} = \emptyset$

  ▷ BiTrajDiff Model Training
  Train the forward and backward trajectory diffusion model $\epsilon_{\theta f}$ and $\epsilon_{\theta b}$ with $\mathcal{D}$ according to Eq. 4.
  Train the inverse dynamics model $f_\psi$ and reward model $r_\phi$ with $\mathcal{D}$ according to Eq. 6
  ▷ BiTrajDiff Data Generation
  **While** $|\mathcal{D}_{\text{da}}| \leq \sigma * |\mathcal{D}|$ **do**
    # Bidirectional State Trajectory Generation
    Sample random batch of states $\mathcal{B} = \{s_t\} \sim \mathcal{D}$.
    Generate $\mathbf{x}_0^f(\tau) = \{\hat{s}_t, \hat{s}_{t+1}, \cdots, \hat{s}_{t+H-1}\}$ with $\epsilon_{\theta f}$ conditioned on $s_t$ and $R^f$.
    Generate $\mathbf{x}_0^b(\tau) = \{\hat{s}_{t-H+1}, \cdots, \hat{s}_{t-1}, \hat{s}_t\}$ with $\epsilon_{\theta b}$ conditioned on $s_t$ and $R^b$.
    Construct the bidirectional state trajectory $\hat{\tau}_s(s_t)$ with $\mathbf{x}_0^b(\tau)$ and $\mathbf{x}_0^f(\tau)$ according to Eq. 5.
    # Bidirectional Trajectory Completion
    Compute the action $\hat{a}_i$ and reward $\hat{r}_i$ signal with $\hat{\tau}_s(s_t)$ by $f_\psi$ and $r_\phi$.
    Obtain the generated trajectory set $\hat{\mathcal{D}}$ containing $\hat{\tau}(s_t) = \{(\hat{s}_i, \hat{a}_i, \hat{r}_i, \hat{s}_{i+1})\}_{i=t-H+1}^{t+H-2}$
    # Bidirectional Trajectory Filter
    Filter the Top-$C_{\text{ood}}$ trajectories with smallest $d(\hat{\tau})$ from $\hat{\mathcal{D}}$
    Filter the Top-$C_{\text{greedy}}$ trajectories as batch $\tilde{\mathcal{D}}$ with largest $R(\hat{\tau})$ from the Top-$C_{\text{ood}}$ trajectories.
    Add the final generated batch into the augmented dataset: $\mathcal{D}_{\text{da}} = \mathcal{D}_{\text{da}} \bigcup \tilde{\mathcal{D}}$
  **End while**

---

## B. Experiments Details

### B.1. Task Abbreviations and Task Versions

For improved readability and conciseness, we use abbreviations for the locomotion tasks throughout the main text, with the corresponding definitions provided in Table S1.

### B.2. Implementation Details

#### B.2.1. BiTrajDiff Implementation

In this section, we provide the implementation details of our experiments. We conducted our experiments on a cluster of 4 A100 GPUs. The source code will be made publicly available upon the publication of this paper. Our BiTrajDiff is implemented based on CleanDiffuser (Dong et al., 2024), a popular modularized library for diffusion models in decision making. We represent the noise model $\epsilon_\theta$ with the 1D DiT backbone (Peebles & Xie, 2022), consisting 2 transformer blocks with adaptive layer normalization. Each block contains a multi-head self-attention layer followed by a feed-forward MLP. The diffusion timestep embeddings are first encoded via Fourier features and then projected through a two-layer MLP network with 128 hidden units. Similarly, the condition inputs are encoded by a two-layer MLP network with 128

*Table S1.* Abbreviations of the corresponding locomotion tasks and datasets.

| Dataset | halfcheetah | walker2d | hopper |
|---|---|---|---|
| medium | hc-m | w-m | h-m |
| medium-replay | hc-m-r | w-m-r | h-m-r |
| medium-expert | hc-m-e | w-m-e | h-m-e |

*Table S2.* Test returns of our proposed BiTrajDiff and baselines on the Gym tasks. $\pm$ corresponds to the standard deviation of the performance on 5 random seeds. The best and the second-best results of each setting are marked as **bold** and underline, respectively.

| Source | Task | CQL (Kumar et al., 2020) | | | | | DT (Chen et al., 2021) | | | | |
|---|---|---|---|---|---|---|---|---|---|---|---|
| | | Base | RTDiff | Synther | DiffStitch | Ours | Base | RTDiff | Synther | DiffStitch | Ours |
| medium | halfcheetah | $47.4_{\pm0.2}$ | $47.8_{\pm1.6}$ | $\underline{48.3}_{\pm0.4}$ | $48.0_{\pm0.2}$ | $\mathbf{48.7}_{\pm0.3}$ | $42.7_{\pm0.5}$ | $42.6_{\pm2.5}$ | $42.7_{\pm0.1}$ | $\underline{42.8}_{\pm0.4}$ | $\mathbf{43.3}_{\pm0.2}$ |
| | walker2d | $82.6_{\pm1.0}$ | $\underline{83.5}_{\pm1.7}$ | $83.4_{\pm0.5}$ | $82.4_{\pm4.5}$ | $\mathbf{84.2}_{\pm0.5}$ | $69.6_{\pm11.0}$ | $71.6_{\pm10.2}$ | $\underline{74.8}_{\pm4.3}$ | $72.3_{\pm9.4}$ | $\mathbf{77.8}_{\pm4.0}$ |
| | hopper | $68.8_{\pm4.1}$ | $70.9_{\pm3.7}$ | $\underline{75.5}_{\pm8.4}$ | $69.7_{\pm5.8}$ | $\mathbf{80.9}_{\pm3.4}$ | $56.0_{\pm1.9}$ | $56.6_{\pm5.1}$ | $\underline{58.1}_{\pm2.4}$ | $\mathbf{58.4}_{\pm2.4}$ | $57.8_{\pm2.2}$ |
| medium expert | halfcheetah | $87.2_{\pm4.6}$ | $90.0_{\pm2.9}$ | $\underline{92.2}_{\pm3.5}$ | $91.4_{\pm5.0}$ | $\mathbf{94.1}_{\pm1.1}$ | $70.9_{\pm12.3}$ | $72.3_{\pm7.0}$ | $\underline{80.4}_{\pm9.8}$ | $76.9_{\pm12.7}$ | $\mathbf{83.5}_{\pm6.3}$ |
| | walker2d | $\underline{110.2}_{\pm0.6}$ | $110.1_{\pm0.8}$ | $110.1_{\pm0.4}$ | $109.4_{\pm3.6}$ | $\mathbf{110.4}_{\pm0.4}$ | $104.6_{\pm6.9}$ | $101.2_{\pm1.8}$ | $\underline{105.7}_{\pm3.5}$ | $104.0_{\pm8.6}$ | $\mathbf{106.3}_{\pm6.0}$ |
| | hopper | $103.5_{\pm8.1}$ | $102.2_{\pm3.2}$ | $103.2_{\pm10.2}$ | $\underline{105.3}_{\pm7.8}$ | $\mathbf{105.7}_{\pm5.8}$ | $107.5_{\pm3.4}$ | $\underline{109.6}_{\pm1.3}$ | $109.2_{\pm0.5}$ | $108.8_{\pm4.8}$ | $\mathbf{110.1}_{\pm1.9}$ |
| medium replay | halfcheetah | $\underline{45.9}_{\pm0.2}$ | $45.6_{\pm0.4}$ | $\mathbf{47.4}_{\pm0.6}$ | $45.1_{\pm0.4}$ | $45.8_{\pm0.3}$ | $38.8_{\pm2.4}$ | $\underline{39.1}_{\pm4.1}$ | $39.3_{\pm1.3}$ | $38.9_{\pm2.6}$ | $\mathbf{39.5}_{\pm0.8}$ |
| | walker2d | $82.1_{\pm1.8}$ | $\underline{84.4}_{\pm3.2}$ | $\mathbf{84.8}_{\pm0.6}$ | $82.4_{\pm1.8}$ | $\mathbf{84.8}_{\pm1.5}$ | $52.1_{\pm5.5}$ | $53.8_{\pm2.4}$ | $\underline{56.1}_{\pm5.2}$ | $54.6_{\pm7.5}$ | $\mathbf{58.2}_{\pm6.1}$ |
| | hopper | $95.7_{\pm2.5}$ | $95.9_{\pm7.1}$ | $96.1_{\pm1.2}$ | $\underline{98.2}_{\pm6.8}$ | $\mathbf{100.3}_{\pm1.3}$ | $67.3_{\pm8.9}$ | $\mathbf{70.2}_{\pm10.6}$ | $69.0_{\pm16.2}$ | $68.1_{\pm19.7}$ | $\underline{69.6}_{\pm13.1}$ |
| **Average** | | 80.4 | 81.2 | 82.3 | 81.3 | **83.9** | 67.7 | 68.6 | 70.6 | 69.4 | **71.8** |

*Table S3.* Test returns of our proposed BiTrajDiff and baselines on Maze and Franka Kitchen tasks.

| Task | CQL (Kumar et al., 2020) | | | | | DT (Chen et al., 2021) | | | | |
|---|---|---|---|---|---|---|---|---|---|---|
| | Base | RTDiff | Synther | DiffStitch | Ours | Base | RTDiff | Synther | DiffStitch | Ours |
| maze2d-umaze | $4.5_{\pm1.1}$ | $\underline{11.3}_{\pm8.3}$ | $2.5_{\pm4.6}$ | $2.3_{\pm1.4}$ | $\mathbf{18.6}_{\pm9.4}$ | $24.9_{\pm10.6}$ | $\underline{29.8}_{\pm14.5}$ | $31.4_{\pm9.5}$ | $22.9_{\pm2.7}$ | $\mathbf{36.1}_{\pm8.7}$ |
| maze2d-medium | $84.4_{\pm24.5}$ | $87.7_{\pm23.2}$ | $\underline{90.5}_{\pm10.7}$ | $72.9_{\pm16.8}$ | $\mathbf{95.8}_{\pm17.4}$ | $16.6_{\pm2.2}$ | $18.8_{\pm2.7}$ | $\underline{24.7}_{\pm3.1}$ | $19.3_{\pm1.6}$ | $\mathbf{26.6}_{\pm4.0}$ |
| maze2d-large | $34.3_{\pm25.5}$ | $38.9_{\pm20.7}$ | $45.3_{\pm14.2}$ | $\mathbf{65.2}_{\pm72.5}$ | $\underline{48.3}_{\pm28.7}$ | $22.3_{\pm12.5}$ | $\underline{24.9}_{\pm4.9}$ | $24.2_{\pm6.7}$ | $24.4_{\pm7.3}$ | $\mathbf{28.5}_{\pm4.4}$ |
| antmaze-umaze-diverse | $31.6_{\pm8.4}$ | $40.6_{\pm9.0}$ | $\underline{42.4}_{\pm6.8}$ | $39.2_{\pm9.3}$ | $\mathbf{48.8}_{\pm4.4}$ | $42.0_{\pm4.9}$ | $\underline{46.4}_{\pm16.7}$ | $44.8_{\pm11.2}$ | $42.6_{\pm4.7}$ | $\mathbf{47.8}_{\pm2.9}$ |
| antmaze-medium-diverse | $57.2_{\pm16.2}$ | $60.4_{\pm12.8}$ | $62.6_{\pm9.8}$ | $\underline{63.6}_{\pm12.6}$ | $\mathbf{73.2}_{\pm11.6}$ | $0.0_{\pm0.0}$ | $\underline{0.2}_{\pm0.4}$ | $0.0_{\pm0.0}$ | $0.0_{\pm0.0}$ | $\mathbf{0.4}_{\pm0.5}$ |
| antmaze-large-diverse | $8.0_{\pm7.3}$ | $5.3_{\pm1.2}$ | $\underline{12.6}_{\pm10.9}$ | $2.6_{\pm3.7}$ | $\mathbf{16.0}_{\pm10.2}$ | $\underline{0.0}_{\pm0.0}$ | $0.0_{\pm0.0}$ | $0.0_{\pm0.0}$ | $\mathbf{0.8}_{\pm0.7}$ | $0.0_{\pm0.0}$ |
| **Maze Mean** | 36.7 | 40.7 | 42.7 | 41.0 | **50.1** | 17.6 | 20.0 | 20.8 | 18.3 | **23.2** |
| kitchen-complete | $7.8_{\pm11.9}$ | $\mathbf{15.2}_{\pm9.3}$ | $8.4_{\pm6.1}$ | $10.5_{\pm5.3}$ | $\underline{13.8}_{\pm7.4}$ | $61.7_{\pm14.6}$ | $61.3_{\pm8.0}$ | $62.3_{\pm8.6}$ | $\underline{63.4}_{\pm11.7}$ | $\mathbf{67.9}_{\pm6.8}$ |
| kitchen-partial | $21.1_{\pm2.5}$ | $\mathbf{25.8}_{\pm3.1}$ | $23.2_{\pm7.3}$ | $19.3_{\pm6.7}$ | $\underline{24.9}_{\pm4.7}$ | $21.6_{\pm11.1}$ | $22.2_{\pm7.5}$ | $21.8_{\pm7.5}$ | $\underline{29.2}_{\pm5.8}$ | $\mathbf{33.5}_{\pm18.9}$ |
| kitchen-mixed | $16.2_{\pm6.4}$ | $19.1_{\pm1.2}$ | $\underline{19.3}_{\pm5.7}$ | $12.0_{\pm9.6}$ | $\mathbf{21.5}_{\pm7.9}$ | $22.5_{\pm16.5}$ | $23.7_{\pm14.7}$ | $\underline{31.6}_{\pm14.8}$ | $24.7_{\pm10.4}$ | $\mathbf{37.0}_{\pm11.7}$ |
| **Kitchen Mean** | 15.0 | 20.0 | 17.0 | 13.9 | **20.1** | 35.3 | 35.7 | 38.6 | 39.1 | **46.1** |

hidden units. For the sampling process, we follow the CleanDiffuser reimplementation of Decision Diffuser (Ajay et al., 2022), employing the VP-SDE (Song et al., 2020) formulation with 20 diffusion steps for sampling. Meanwhile, the inverse dynamics and rewards model are both represented by a two-layer MLP network with 512 hidden units, respectively. During BiTrajDiff training, we employ an Adam optimizer with a learning rate of $2e-4$ for all networks, using a batch size of 64 and performing $1e6$ gradient steps in total. The default trajectory horizon $H$ for both training and generation is set to 5. The ratio $\sigma$ of the augmented dataset is 30%. During our BiTrajDiff data augmentation pipeline, the batch size is set to 512, while the filter number $C_{\text{ood}}$ is 256 and $C_{\text{greedy}}$ is 64. As for the target return-to-go $R^f$ and $R^b$, we set them to be equal in our BiTrajDiff conditional generation. Each task is assigned its own target return, and in our experiments we directly adopt the corresponding hyperparameters from the CleanDiffuser reimplementation of Decision Diffuser (Ajay et al., 2022).

### B.2.2. OFFLINE RL ALGORITHM IMPLEMENTATION

In our experiments, we evaluate BiTrajDiff against four offline RL algorithms: IQL, TD3BC, CQL, and DT. To enable efficient and large-scale evaluation, we conduct all offline RL experiments using the JAX-based library JAX-CORL (Nishimori, 2024). **For fairness, we report re-run results on JAX-CORL with initial datasets as baseline performance, rather than the scores reported in the original papers, to account for discrepancies across codebases.** Meanwhile, for each offline RL algorithm, we ensure that the hyperparameters are configured consistently with those reported in the respective original papers.

## C. Results on CQL and DT

**Results for Locomotion Tasks.** The results on D4RL locomotion tasks are shown in Table S2. BiTrajDiff achieves the highest average returns in all of the environments under both CQL and DT baselines, with improvements of 4.4% and 6.1%, respectively. At the same time, BiTrajDiff maintains high stability across all locomotion tasks, substantially exceeding other tested DA methods. These results highlight the effectiveness of BiTrajDiff.

**Results for Maze and Franka Kitchen Tasks.** Table S3 shows the experimental results on the D4RL Maze and Franka Kitchen tasks. BiTrajDiff acquires the highest average return across all tested tasks of sparse reward scenarios under both CQL and DT baselines. Meanwhile, BiTrajDiff maintains higher stability in performance improvements across all the settings than other DA methods like RTDiff. This observation confirms that BiTrajDiff can synthesize novel and high-quality long-horizon trajectories by stitching forward-future and backward-history paths to address the challenge of the availability of high-quality data under sparse-reward settings. This result underlines the efficacy and applicability of BiTrajDiff in enhancing data diversity and quality.

## D. Extended Discussion on Trajectory Stitching Methods

**Related Work Discussion.** Recent trajectory stitching methods for offline reinforcement learning (Liu et al., 2024c; Zhou et al., 2025; Liu et al., 2024b; Xia et al., 2024) can be broadly categorized into two main paradigms. The first paradigm is *model-based stitching*, where learned dynamics models are used to generate trajectories step by step in order to connect separated states or sub-trajectories. Representative methods include MBRCSL (Zhou et al., 2024) and ASTRO (Yu et al., 2025). This paradigm utilizes a learned dynamic model to naturely connect distant states by rollouts and thereby recover long-range trajectory-level connectivity absent from the logged dataset. However, this flexibility also introduces error propagation due to auto-regressive rollout . Consequently, although model-based stitching can expand the apparent support of the dataset, the generated bridge may suffer from degraded local dynamic validity, especially near precise stitching points or in regions where the learned model is extrapolating beyond well-supported transitions. The second paradigm is *direct middle-segment generation*, where the method either directly synthesizes intermediate trajectory segments between endpoints. DiffStitch (Li et al., 2024), SSD (Kim et al., 2024), SCoTS (Lee & Choi, 2026), and CD/CompDiffuser (Luo et al., 2026) are representative examples of this direction. By explicitly modeling missing middle segments, these methods can improve goal-reaching behavior, trajectory diversity, and compositional generalization. Nevertheless, strict dynamic consistency at the exact stitching points remains an inherent challenge. The generated segment must be simultaneously compatible with the incoming endpoint, the outgoing endpoint, and the environment dynamics; when the offline data provide only sparse evidence for such a connection, satisfying all three constraints becomes difficult even if the global trajectory appears semantically plausible.

**Mechanistic Advantage of BiTrajDiff.** Instead of relying on single-direction autoregressive rollouts or unconstrained middle-segment generation,BiTrajDiff uses an anchor-centered factorization to recover missing trajectory-level couplings. Given an anchor state, it independently generates a backward-history segment and a forward-future segment, and stitches them at the shared anchor $s_t$. This factorization is an operational design principle rather than an assumption of true conditional independence. It enables BiTrajDiff to compose locally supported incoming and outgoing behavior patterns whose pairing may be absent in the dataset, while avoiding long autoregressive rollouts that accumulate prediction errors. The completed trajectories are further screened by the OOD and greedy filters, which respectively discourage unsupported transitions and retain high-return candidates. Thus, BiTrajDiff expands trajectory-level support while preserving local dynamic consistency.

**Empirical Validation.** The empirical comparisons further support this mechanistic interpretation. Table S4 evaluates the stitching and compositional connectivity recovery ability of BiTrajDiff on selected OGBench (Park et al., 2025a) single-task offline RL problems, where the compared baselines include the single-direction method SynthER (Lu et al., 2023) and representative stitching methods DiffStitch (Li et al., 2024), ASTRO (Yu et al., 2025), and SSD (Kim et al., 2024). Under both IQL and FQL backbones, BiTrajDiff achieves the strongest average performance, indicating that the shared-anchor bidirectional generation mechanism more effectively recovers missing trajectory-level connectivity than methods that rely on single-direction generation, autoregressive rollout, or direct segment stitching alone. Table S5 further extends this comparison to goal-conditioned offline RL, where BiTrajDiff is evaluated against SynthER, SCoTS (Lee & Choi, 2026), and CD/CompDiffuser (Luo et al., 2026). Its leading average performance under both GCIQL and CRL shows that the same mechanism remains effective when the policy must compose trajectories toward explicit goals in stitching and exploration

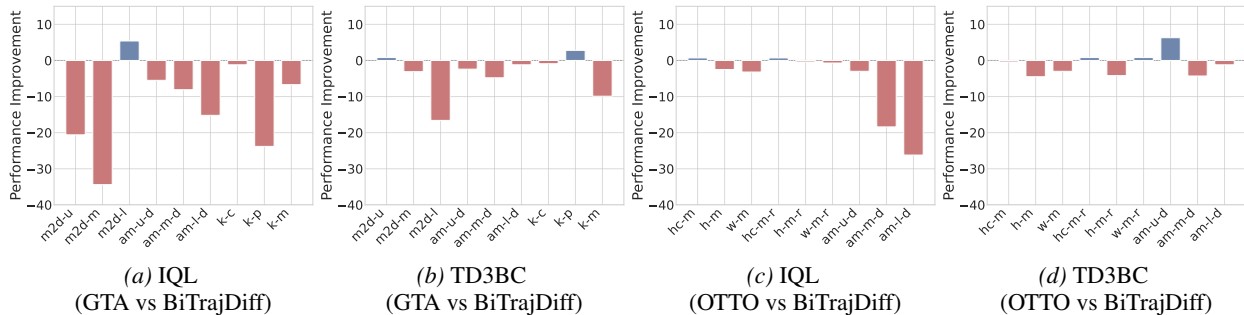

*Figure S1.* Performance enhancement comparison with GTA (Lee et al., 2024) and OTTO (Zhao et al., 2025) on IQL and TD3BC.

*Table S4.* Offline RL performance comparison under our proposed BiTrajDiff and baselines on selected single-task OGBench tasks.

| Task | IQL (Kostrikov et al., 2021) | | | | | | FQL (Park et al., 2025b) | | | | | |
|---|---|---|---|---|---|---|---|---|---|---|---|---|
| | Base | ASTRO | DiffStitch | Synther | SSD | Ours | Base | ASTRO | DiffStitch | Synther | SSD | Ours |
| antmaze-large-stitch-singletask-task1 | 26.2 | 51.7 | 35.0 | 31.1 | $53_{\pm7}$ | $\mathbf{64}_{\pm10}$ | 29.2 | 57.3 | 33.1 | 28.7 | $75_{\pm22}$ | $\mathbf{79}_{\pm13}$ |
| humanoidmaze-medium-stitch-singletask-task1 | 29.7 | 31.4 | 28.3 | 31.2 | $33_{\pm11}$ | $\mathbf{38}_{\pm7}$ | 17.5 | 30.0 | 22.6 | 15.9 | $43_{\pm16}$ | $\mathbf{47}_{\pm18}$ |
| cube-single-play-singletask-task1 | 81.5 | $\mathbf{89.2}$ | 79.0 | 82.4 | $82_{\pm7}$ | $87_{\pm5}$ | 88.0 | 92.9 | 89.6 | 87.3 | $93_{\pm4}$ | $\mathbf{98}_{\pm4}$ |
| **Average** | 45.8 | 57.4 | 47.4 | 48.2 | 56.0 | **63.0** | 44.9 | 60.1 | 48.4 | 44.0 | 70.3 | **74.7** |

*Table S5.* Goal-conditioned offline RL performance comparison under our proposed BiTrajDiff and baselines on OGBench tasks.

| Task | GCIQL (Kostrikov et al., 2021) | | | | | CRL (Kumar et al., 2020) | | | | |
|---|---|---|---|---|---|---|---|---|---|---|
| | Base | Synther | SCoTS | CD | Ours | Base | Synther | SCoTS | CD | Ours |
| antmaze-medium-stitch-v0 | $29_{\pm6}$ | $31_{\pm3}$ | $35_{\pm2}$ | $34_{\pm4}$ | $\mathbf{38}_{\pm2}$ | $53_{\pm6}$ | $48_{\pm3}$ | $65_{\pm3}$ | $61_{\pm4}$ | $\mathbf{68}_{\pm9}$ |
| antmaze-large-stitch-v0 | $7_{\pm2}$ | $3_{\pm4}$ | $7_{\pm1}$ | $7_{\pm3}$ | $\mathbf{8}_{\pm4}$ | $11_{\pm2}$ | $12_{\pm2}$ | $19_{\pm1}$ | $17_{\pm2}$ | $\mathbf{19}_{\pm2}$ |
| antmaze-medium-explore-v0 | $13_{\pm2}$ | $12_{\pm3}$ | $18_{\pm3}$ | $17_{\pm5}$ | $\mathbf{19}_{\pm4}$ | $3_{\pm2}$ | $3_{\pm1}$ | $15_{\pm3}$ | $11_{\pm2}$ | $14_{\pm2}$ |
| **Average** | 16.3 | 15.3 | 20.0 | 19.3 | **21.7** | 22.3 | 21.0 | 33.0 | 29.7 | **33.7** |

environments. Finally, Figure S1 compares BiTrajDiff with two additional DA baselines, the diffusion-based method GTA (Lee et al., 2024) and the world-model-based method OTTO (Zhao et al., 2025), under the IQL and TD3BC backbones. Although GTA and OTTO can improve performance in some settings, their gains are generally smaller and less consistent, whereas BiTrajDiff achieves more robust improvements by synthesizing useful long-horizon transitions while controlling trajectory inconsistency through completion and filtering. Taken together, Table S4, Table S5, and Figure S1 show that effective trajectory stitching requires both support expansion and dynamic consistency: the augmented data should introduce new connections between behavior fragments, which remain locally compatible with the environment dynamics.

## E. Ablation Study on the Direction of Diffusion Generation

We conduct experiments with different augmented diffusion trajectories generated in single-directional and bi-directional. Table S6 shows the test returns of the IQL and TD3BC algorithms. For CQL and DT algorithms, the quantitative results are reported in Table S7. The algorithms were evaluated on D4RL locomotion tasks with augmented trajectories generated in single-direction and bi-directional. Compared to the base method, the forward method offers only modest gains and is at times even detrimental to performance. Conversely, the backward method generally outperforms the base method across most tasks, achieving substantial gains. Finally, our bi-directional approach, BiTrajDiff, consistently exhibits superior and robust performance. Our BiTrajDiff enables global connectivity between originally unreachable states from the original dataset and thus constructs novel and high-quality augmented trajectories.

*Table S6.* Test returns of our proposed BiTrajDiff with augmented single- and bi-directional diffusion generated trajectories on IQL and TD3BC algorithms under D4RL locomotion tasks.

| Source | Task | IQL (Kostrikov et al., 2021) | | | | TD3BC (Fujimoto & Gu, 2021) | | | |
|---|---|---|---|---|---|---|---|---|---|
| | | Base | Forward | Backward | Ours | Base | Forward | Backward | Ours |
| medium | halfcheetah | $48.2_{\pm0.2}$ | $47.3_{\pm0.8}$ | $\underline{48.4}_{\pm0.5}$ | $\mathbf{48.6}_{\pm0.2}$ | $48.4_{\pm0.2}$ | $50.0_{\pm0.3}$ | $\mathbf{50.8}_{\pm0.7}$ | $\underline{50.3}_{\pm0.4}$ |
| | walker2d | $77.6_{\pm5.3}$ | $79.1_{\pm4.9}$ | $\underline{83.7}_{\pm2.6}$ | $\mathbf{86.7}_{\pm1.7}$ | $82.0_{\pm2.7}$ | $81.7_{\pm0.7}$ | $\underline{84.6}_{\pm0.4}$ | $\mathbf{86.7}_{\pm1.0}$ |
| | hopper | $67.0_{\pm1.9}$ | $70.2_{\pm7.6}$ | $\underline{75.8}_{\pm4.8}$ | $\mathbf{81.1}_{\pm5.0}$ | $60.4_{\pm3.7}$ | $65.2_{\pm6.7}$ | $\mathbf{80.3}_{\pm4.8}$ | $\underline{79.0}_{\pm3.5}$ |
| medium expert | halfcheetah | $90.1_{\pm4.4}$ | $92.5_{\pm1.2}$ | $\underline{94.2}_{\pm0.1}$ | $\mathbf{95.3}_{\pm0.1}$ | $92.3_{\pm5.1}$ | $94.7_{\pm1.9}$ | $\underline{95.3}_{\pm0.9}$ | $\mathbf{96.3}_{\pm0.4}$ |
| | walker2d | $\underline{112.4}_{\pm0.4}$ | $109.7_{\pm1.6}$ | $\mathbf{112.5}_{\pm0.4}$ | $\mathbf{112.5}_{\pm0.5}$ | $109.3_{\pm0.2}$ | $109.1_{\pm0.5}$ | $\underline{110.0}_{\pm0.4}$ | $\mathbf{110.2}_{\pm0.3}$ |
| | hopper | $105.4_{\pm6.3}$ | $102.3_{\pm7.0}$ | $\underline{109.1}_{\pm5.2}$ | $\mathbf{110.9}_{\pm0.6}$ | $94.2_{\pm11.1}$ | $88.7_{\pm5.5}$ | $\underline{108.4}_{\pm2.5}$ | $\mathbf{109.0}_{\pm5.1}$ |
| medium replay | halfcheetah | $43.5_{\pm0.2}$ | $42.5_{\pm0.8}$ | $\underline{43.7}_{\pm0.1}$ | $\mathbf{44.1}_{\pm0.2}$ | $44.1_{\pm0.1}$ | $44.4_{\pm0.3}$ | $\mathbf{45.1}_{\pm0.5}$ | $\underline{45.0}_{\pm0.6}$ |
| | walker2d | $73.3_{\pm7.5}$ | $79.4_{\pm5.0}$ | $\underline{82.2}_{\pm2.9}$ | $\mathbf{87.6}_{\pm2.2}$ | $80.9_{\pm6.0}$ | $\underline{87.2}_{\pm2.4}$ | $85.8_{\pm4.9}$ | $\mathbf{89.9}_{\pm1.4}$ |
| | hopper | $94.7_{\pm4.0}$ | $\underline{97.2}_{\pm3.0}$ | $89.6_{\pm3.1}$ | $\mathbf{102.8}_{\pm0.6}$ | $64.4_{\pm8.9}$ | $83.5_{\pm12.9}$ | $\underline{83.8}_{\pm9.7}$ | $\mathbf{85.0}_{\pm10.3}$ |
| **Average** | | 79.1 | 80.0 | 82.1 | **85.5** | 75.1 | 78.3 | 82.7 | **83.5** |

*Table S7.* Test returns of our proposed BiTrajDiff with augmented single- and bi-directional diffusion generated trajectories on CQL and DT algorithms under D4RL locomotion tasks.

| Source | Task | CQL (Kumar et al., 2020) | | | | DT (Chen et al., 2021) | | | |
|---|---|---|---|---|---|---|---|---|---|
| | | Base | Forward | Backward | Ours | Base | Forward | Backward | Ours |
| medium | halfcheetah | $47.4_{\pm0.2}$ | $47.5_{\pm0.2}$ | $\mathbf{48.9}_{\pm0.4}$ | $\underline{48.7}_{\pm0.3}$ | $42.7_{\pm0.5}$ | $42.7_{\pm0.2}$ | $\underline{43.0}_{\pm0.3}$ | $\mathbf{43.3}_{\pm0.2}$ |
| | walker2d | $82.6_{\pm1.0}$ | $80.9_{\pm2.7}$ | $\underline{83.8}_{\pm0.7}$ | $\mathbf{84.2}_{\pm0.5}$ | $69.6_{\pm11.0}$ | $74.6_{\pm3.6}$ | $\underline{76.1}_{\pm3.5}$ | $\mathbf{77.8}_{\pm4.0}$ |
| | hopper | $68.8_{\pm4.1}$ | $71.6_{\pm14.9}$ | $\underline{75.2}_{\pm9.5}$ | $\mathbf{80.9}_{\pm3.4}$ | $56.0_{\pm1.9}$ | $56.6_{\pm5.6}$ | $\underline{56.9}_{\pm5.7}$ | $\mathbf{57.8}_{\pm2.2}$ |
| medium expert | halfcheetah | $87.2_{\pm4.6}$ | $88.0_{\pm3.9}$ | $\underline{92.6}_{\pm2.2}$ | $\mathbf{94.1}_{\pm1.1}$ | $70.8_{\pm12.3}$ | $\underline{74.3}_{\pm4.7}$ | $64.7_{\pm4.4}$ | $\mathbf{83.5}_{\pm6.3}$ |
| | walker2d | $110.2_{\pm0.6}$ | $\mathbf{110.5}_{\pm0.6}$ | $109.5_{\pm0.7}$ | $\underline{110.4}_{\pm0.4}$ | $104.6_{\pm6.9}$ | $\underline{105.1}_{\pm5.1}$ | $104.9_{\pm4.8}$ | $\mathbf{106.3}_{\pm6.0}$ |
| | hopper | $\underline{103.5}_{\pm8.1}$ | $102.7_{\pm13.9}$ | $102.5_{\pm7.8}$ | $\mathbf{105.7}_{\pm5.8}$ | $107.5_{\pm3.4}$ | $108.3_{\pm3.3}$ | $\underline{109.8}_{\pm2.4}$ | $\mathbf{110.1}_{\pm1.9}$ |
| medium replay | halfcheetah | $\underline{45.9}_{\pm0.2}$ | $\underline{45.9}_{\pm0.5}$ | $\mathbf{46.2}_{\pm0.4}$ | $45.8_{\pm0.3}$ | $38.8_{\pm2.4}$ | $36.4_{\pm2.9}$ | $\underline{39.3}_{\pm0.5}$ | $\mathbf{39.5}_{\pm0.8}$ |
| | walker2d | $82.1_{\pm1.8}$ | $82.9_{\pm0.6}$ | $\underline{84.4}_{\pm2.1}$ | $\mathbf{84.8}_{\pm1.5}$ | $52.1_{\pm5.5}$ | $\mathbf{58.6}_{\pm10.0}$ | $54.9_{\pm4.2}$ | $\underline{58.2}_{\pm6.1}$ |
| | hopper | $95.7_{\pm2.5}$ | $\underline{97.9}_{\pm6.1}$ | $93.8_{\pm5.3}$ | $\mathbf{100.3}_{\pm1.3}$ | $67.3_{\pm8.9}$ | $62.1_{\pm13.0}$ | $\underline{68.4}_{\pm17.2}$ | $\mathbf{69.6}_{\pm13.1}$ |
| **Average** | | 80.4 | 80.9 | 81.9 | **83.9** | 67.7 | 68.7 | 68.7 | **71.8** |

*Table S8.* Test results of BiTrajDiff with data from different variants related to trajectory filters. OOD indicates the OOD filter, and G represents the greedy filter.

| Source | Task | IQL (Kostrikov et al., 2021) | | | | | TD3BC (Fujimoto & Gu, 2021) | | | | |
|---|---|---|---|---|---|---|---|---|---|---|---|
| | | Base | None | w G, w/o OOD | w OOD, w/o G | Ours | Base | None | w G, w/o OOD | w OOD, w/o G | Ours |
| medium replay | walker2d | $73.3_{\pm7.5}$ | $78.5_{\pm10.2}$ | $\underline{83.9}_{\pm4.5}$ | $80.1_{\pm7.0}$ | $\mathbf{87.6}_{\pm2.2}$ | $80.9_{\pm6.0}$ | $77.2_{\pm24.4}$ | $77.5_{\pm26.9}$ | $\underline{82.5}_{\pm8.0}$ | $\mathbf{89.9}_{\pm1.4}$ |
| | hopper | $94.7_{\pm4.0}$ | $93.6_{\pm12.0}$ | $91.6_{\pm8.2}$ | $\underline{96.3}_{\pm3.5}$ | $\mathbf{102.8}_{\pm0.6}$ | $64.4_{\pm8.9}$ | $69.0_{\pm19.2}$ | $\underline{75.0}_{\pm17.0}$ | $74.6_{\pm23.6}$ | $\mathbf{85.0}_{\pm10.3}$ |

*Table S9.* Dynamic error $\mathcal{E}_{\text{Dyn}}$ and L2 distance $\mathcal{E}_{\text{L2D}}$ comparison of trajectories generated by BiTrajDiff with different filtering strategies.

| $\mathcal{E}_{\text{Dyn}}\downarrow/\mathcal{E}_{\text{L2D}}\uparrow$ | w/o OOD + Greedy | w OOD, w/o Greedy | w Greedy, w/o OOD | w OOD + Greedy Filter (Ours) |
|---|---|---|---|---|
| halfcheetah-m-e | 0.29 / 9.39 | 0.15 / 8.73 | 0.36 / 11.27 | 0.18 / 10.45 |
| hopper-m-e | 0.44 / 15.04 | 0.23 / 15.61 | 0.71 / 18.58 | 0.30 / 16.89 |
| walker2d-m-e | 0.34 / 11.26 | 0.22 / 10.36 | 0.47 / 14.97 | 0.26 / 14.10 |

*Table S10.* Performance comparison of BiTrajDiff with different augmented data ratios $\sigma$ on the *walker2d-medium-replay*. The best result for each offline RL algorithm is marked as **bold**.

| Ratio | IQL | TD3BC | CQL |
|---|---|---|---|
| 0% | $73.3_{\pm7.5}$ | $80.9_{\pm6.0}$ | $82.1_{\pm1.8}$ |
| 10% | $78.7_{\pm2.4}$ | $82.0_{\pm5.5}$ | $84.0_{\pm3.1}$ |
| 20% | $84.2_{\pm1.3}$ | $87.5_{\pm1.0}$ | $83.9_{\pm2.0}$ |
| 30% | $87.6_{\pm2.2}$ | $\mathbf{89.9}_{\pm1.4}$ | $84.8_{\pm1.5}$ |
| 40% | $\mathbf{88.7}_{\pm4.5}$ | $86.7_{\pm2.9}$ | $84.7_{\pm2.1}$ |
| 50% | $81.6_{\pm5.6}$ | $86.4_{\pm2.7}$ | $\mathbf{86.3}_{\pm2.1}$ |
| 100% | $84.5_{\pm10.2}$ | $86.7_{\pm2.4}$ | $83.9_{\pm3.3}$ |

*Table S11.* Test returns of our proposed BiTrajDiff and baselines with varying length of $n$-step TD estimator ($n$) on *walker2d-medium-replay*.

| Length of n-step TD Estimator($n$) | IQL (Kostrikov et al., 2021) | | | | TD3BC (Fujimoto & Gu, 2021) | | | |
|---|---|---|---|---|---|---|---|---|
| | base | RTDiff | diff stitch | ours | base | RTDiff | diff stitch | ours |
| $n=2$ | $73.3_{\pm7.5}$ | $74.0_{\pm5.2}$ | $\underline{74.9}_{\pm8.2}$ | $\mathbf{87.6}_{\pm2.2}$ | $80.9_{\pm6.0}$ | $\underline{83.5}_{\pm7.4}$ | $80.5_{\pm5.2}$ | $\mathbf{89.9}_{\pm1.4}$ |
| $n=5$ | $71.7_{\pm12.4}$ | $76.4_{\pm16.6}$ | $\underline{79.8}_{\pm2.4}$ | $\mathbf{86.7}_{\pm0.9}$ | $80.3_{\pm12.0}$ | $\mathbf{88.6}_{\pm3.6}$ | $87.7_{\pm4.0}$ | $87.3_{\pm1.3}$ |
| $n=10$ | $69.5_{\pm13.5}$ | $\underline{73.1}_{\pm12.7}$ | $71.4_{\pm9.9}$ | $\mathbf{79.5}_{\pm2.3}$ | $\underline{81.5}_{\pm7.6}$ | $79.6_{\pm12.7}$ | $81.3_{\pm5.1}$ | $\mathbf{87.5}_{\pm3.4}$ |
| $n=15$ | $\underline{72.7}_{\pm11.4}$ | $67.2_{\pm14.8}$ | $69.7_{\pm6.0}$ | $\mathbf{77.9}_{\pm6.2}$ | $75.1_{\pm9.9}$ | $\underline{80.6}_{\pm13.8}$ | $79.1_{\pm9.7}$ | $\mathbf{83.9}_{\pm4.7}$ |

# F. Effectiveness of Trajectory Filters

We illustrate the necessity of integrating both OOD and greedy filters in our full BiTrajDiff model. Without any filters, the performance is comparable to, and in some cases inferior to, that of the base method on both the IQL and TD3BC algorithms. When applying the greedy filter in isolation, the performance improvement is noticeable but marred by volatility. In contrast, using the OOD filter alone yields a more stable and substantial performance enhancement. Nevertheless, our full BiTrajDiff model, which integrates both filters, consistently outperforms all other configurations across the tested settings, indicating the necessity of both OOD and greedy filters.

To further investigate the mechanism of our trajectory filters, we quantitatively measure the dynamic error $\mathcal{E}_{\text{Dyn}}$ and L2 distance $\mathcal{E}_{\text{L2D}}$ of trajectories generated by BiTrajDiff with different filter combinations, as shown in Table S9. The results show that removing both filters leads to larger dynamic errors, indicating that unfiltered generated trajectories can contain inconsistent transitions. Using only the OOD filter substantially reduces dynamic error, but also limits trajectory diversity, as reflected by smaller L2 distances. Using only the greedy filter increases L2 distance, but introduces much larger dynamic errors. In contrast, BiTrajDiff with both OOD and greedy filters achieves a better balance: it maintains comparable dynamic consistency while preserving larger trajectory-level novelty.

# G. Ablation Study on Augmented Data Ratio $\sigma$

In this section, we conduct experiments to assess the impact of the augmented data ratio $\sigma$. As shown in Table S10, when the ratio $\sigma$ is too low, which is between $0\% \sim 20\%$, performance improvements remain consistently constrained. Conversely, while an augmentation ratio $\sigma$ between 30% and 50% already provides a substantial and stable performance improvement, increasing $\sigma$ to 100% compromises this stability, indicating a decline in the performance. This result demonstrates that both the limited and overabundant synthetic data have detrimental effects on the performance of BiTrajDiff, since the limited data is not enough to provide offline RL algorithms with new behavior patterns, and overabundant data introduces excessive noise, which hinders offline RL algorithms from learning effectively, and an optimal trade-off is achieved with a 30% to 50% augmentation range. As a result, we choose $\sigma = 30\%$ as the fixed hyperparameter for all our experiments.

# H. Performance under $n$-step TD Estimator

In this section, we demonstrate the effectiveness of BiTrajDiff in Offline RL algorithms with varying lengths of the $n$-step TD estimators. As shown in Table S11 and Table S12, while RTDiff and diffsitch initially outperform the base method,

*Table S12.* Test returns of our proposed BiTrajDiff and baselines with varying length of $n$-step TD estimator ($n$) on *hopper-medium-replay*.

| Length of n-step TD Estimator($n$) | IQL (Kostrikov et al., 2021) | | | | TD3BC (Fujimoto & Gu, 2021) | | | |
|:---:|:---:|:---:|:---:|:---:|:---:|:---:|:---:|:---:|
| | base | RTDiff | diff stitch | ours | base | RTDiff | diff stitch | ours |
| $n = 2$ | $94.7_{\pm4.0}$ | $\underline{97.7}_{\pm2.7}$ | $96.4_{\pm5.6}$ | $\mathbf{102.8}_{\pm0.6}$ | $64.4_{\pm8.9}$ | $\underline{72.1}_{\pm18.9}$ | $71.5_{\pm21.0}$ | $\mathbf{85.0}_{\pm10.3}$ |
| $n = 5$ | $90.5_{\pm1.9}$ | $94.7_{\pm5.4}$ | $\mathbf{99.9}_{\pm6.3}$ | $\underline{98.7}_{\pm6.4}$ | $67.0_{\pm26.6}$ | $\underline{75.3}_{\pm14.8}$ | $67.7_{\pm21.0}$ | $\mathbf{89.7}_{\pm12.6}$ |
| $n = 10$ | $94.5_{\pm9.8}$ | $\underline{97.0}_{\pm2.0}$ | $93.3_{\pm9.1}$ | $\mathbf{100.0}_{\pm1.3}$ | $62.7_{\pm25.5}$ | $\underline{66.6}_{\pm15.6}$ | $63.3_{\pm25.7}$ | $\mathbf{83.5}_{\pm13.5}$ |
| $n = 15$ | $82.0_{\pm26.9}$ | $\underline{93.3}_{\pm9.1}$ | $88.4_{\pm9.5}$ | $\mathbf{99.5}_{\pm6.8}$ | $\underline{65.0}_{\pm26.0}$ | $64.8_{\pm25.7}$ | $39.6_{\pm17.2}$ | $\mathbf{71.3}_{\pm14.3}$ |

their performances exhibit a sharp decline, even being surpassed by the base method, as $n$ increases. Conversely, our proposed BiTrajDiff generally achieves the highest scores and consistently maintains its advantage as the $n$-step horizon increases. Meanwhile, as described in Section 5.4, the performance of all tested methods degrades with an increasing $n$-step horizon, which is attributed to the escalating variance in value estimation. Moreover, for DA methods, the accumulated errors introduced by the inverse dynamics model should also be considered. In contrast, BiTrajDiff alleviates these issues by synthesizing novel, high-quality long-horizon trajectories, thereby achieving a more stable and consistent performance. This result demonstrates that by stitching forward-future and backward-history trajectories, BiTrajDiff synthesizes higher-quality and more reliable trajectories than existing DA methods.

## I. Ablation Study on Horizon Lengths

We further investigate the influence of the generation horizon length $H$ in BiTrajDiff on *walker2d-medium-replay-v2*. Since BiTrajDiff performs bidirectional generation around the anchor state, the effective bridge span after stitching becomes $2H - 2$ transitions. The corresponding results are shown in Figure S2. The results show that a moderate horizon length provides the best trade-off between trajectory diversity and dynamics reliability. Specifically, increasing the horizon from $H = 3$ to $H = 5$ substantially improves the downstream Offline RL performance under both IQL and TD3BC, indicating that moderate bidirectional bridge generation helps recover additional trajectory-level connectivity absent from the original dataset. However, further increasing the horizon gradually degrades performance, and excessively long horizons such as $H = 20$ lead to severe instability. This trend is also consistent with the observations reported in RTDiff. Intuitively, longer generated segments are harder to model accurately, resulting in accumulated modeling errors in the synthesized

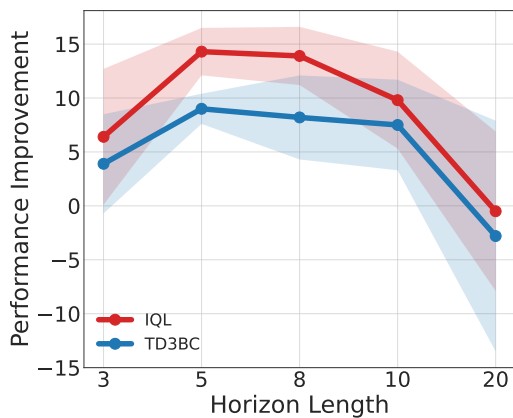

*Figure S2.* Ablation study on $H$.

trajectories, which further propagate through downstream TD estimation and value learning. Therefore, although larger bridge spans may introduce richer connectivity patterns, overly long horizons reduce the reliability of the generated data and ultimately harm offline policy optimization.

## J. Time Overhead

We measure the runtime overhead during both the training and generation stages of BiTrajDiff and other diffusion-based DA methods, as shown in Figure S3. The comparison is conducted on a single 80GB A100 GPU under the *halfcheetah-medium-replay-v2* setting. The results show that BiTrajDiff maintains training and generation runtimes within a comparable range to other diffusion-based DA baselines.

## K. Theoretical Intuition for Bidirectional Stitching

BiTrajDiff is motivated by the trajectory-level connectivity gap between signle-direction and bidirectional trajectory modeling paridgm. Consider an offline dataset $\mathcal{D}$ consisting of trajectories $\tau \in \mathcal{D}$. For a given anchor state $s$, let $h^-$ denote a local history segment arriving at $s$, and let $h^+$ denote a local future segment departing from $s$. We define the incoming and

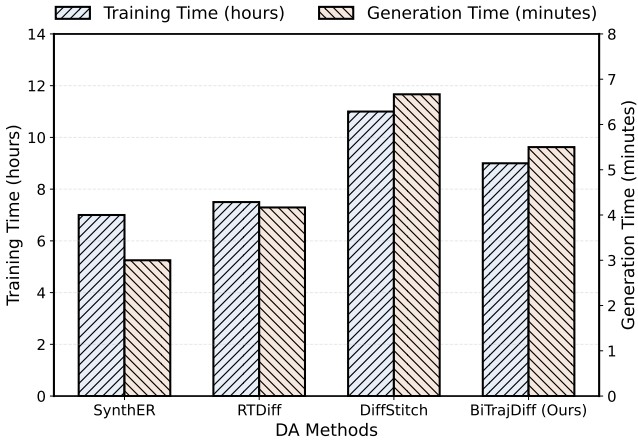

*Figure S3.* Runtime comparison of BiTrajDiff and other DA methods.

outgoing local pattern sets around $s$ as

$$\mathcal{H}_{\mathcal{D}}(s) = \{h^- : \exists \tau \in \mathcal{D}, \ (h^-, s) \subset \tau\}, \tag{7}$$

$$\mathcal{F}_{\mathcal{D}}(s) = \{h^+ : \exists \tau \in \mathcal{D}, \ (s, h^+) \subset \tau\}. \tag{8}$$

Although $\mathcal{H}_{\mathcal{D}}(s)$ and $\mathcal{F}_{\mathcal{D}}(s)$ may both be well supported locally, the dataset only observes a subset of their trajectory-level couplings:

$$\mathcal{C}_{\mathcal{D}}(s) = \{(h^-, h^+) : \exists \tau \in \mathcal{D}, \ (h^-, s, h^+) \subset \tau\} \subseteq \mathcal{H}_{\mathcal{D}}(s) \times \mathcal{F}_{\mathcal{D}}(s). \tag{9}$$

In many offline datasets, $\mathcal{C}_{\mathcal{D}}(s)$ can be much sparser than the full product space $\mathcal{H}_{\mathcal{D}}(s) \times \mathcal{F}_{\mathcal{D}}(s)$. **Therefore, the missing support is not necessarily missing local state coverage, but rather missing support in the joint connectivity structure between incoming and outgoing trajectory patterns.**

The cross-door setting as shown in Figure 1 provides a controlled example of this phenomenon. Doorway-adjacent states $s_d$ can have sufficient one-sided support, meaning that both $\mathcal{H}_{\mathcal{D}}(s_d)$ and $\mathcal{F}_{\mathcal{D}}(s_d)$ contain plausible local segments. However, the observed coupling set $\mathcal{C}_{\mathcal{D}}(s_d)$ may occupy only a small portion of $\mathcal{H}_{\mathcal{D}}(s_d) \times \mathcal{F}_{\mathcal{D}}(s_d)$. This makes the task suitable for trajectory-level stitching: success depends less on extrapolating to unseen local states and more on recovering unseen pairings between locally supported behavior patterns.

This perspective also explains why single-direction data augmentation can be limited in such settings. A single-direction generator is trained on continuations sampled from trajectories in $\mathcal{D}$, so its supervision is concentrated on the observed coupling structure $\mathcal{C}_{\mathcal{D}}(s)$. If a cross-pattern pairing $(h^-, h^+)$ is absent from $\mathcal{D}$, it receives little direct training signal, and the learned generator tends to preserve the empirical connectivity structure rather than create new pairings. In contrast, BiTrajDiff factorizes bridge generation around the same anchor into two conditional generation problems: a backward-history model for $h^-$ and a forward-future model for $h^+$. This factorization allows candidate pairings to be drawn from the richer product space induced by $\mathcal{H}_{\mathcal{D}}(s) \times \mathcal{F}_{\mathcal{D}}(s)$ instead of being restricted to $\mathcal{C}_{\mathcal{D}}(s)$. The inverse dynamics completion and trajectory filters then retain feasible and high-quality candidates. This provides the main theoretical intuition for why bidirectional generation is effective at recovering missing trajectory-level connectivity than other single-direction methods.

