# OpenReview forum: "BiTrajDiff: Bidirectional Trajectory Generation with Diffusion Models for Offline Reinforcement Learning"
_ICML.cc/2026/Conference — ICML 2026 regular_

### Official Review · Reviewer_AYYz · 2026-02-24

**Soundness:** 3
**Presentation:** 3
**Significance:** 3
**Originality:** 3
**Overall Recommendation:** 4
**Confidence:** 4

**Summary:**

summary:
In this paper, the authors proposed a novel BiTrajDiff framework to deal with the hard trajectory stitching problem. Experimental results are provided to show the effectiveness. However, major issues exist, which makes the draft hard to meet the standard of ICML this time.

**Compliance With Llm Reviewing Policy:**

Affirmed.

**Final Justification:**

See my comments.

**Key Questions For Authors:**

Please see weakness.

**Limitations:**

No, please refer to the weakness.

**Strengths And Weaknesses:**

strength:
The problem raised in this work is interesting and important. The paper is easy to follow.

weakness:

1. Omission of recent relevant literature: Several important and highly relevant works concerning trajectory stitching are missing from the current discussion. For example:

[1] State-Covering Trajectory Stitching for Diffusion Planners (NeurIPS 2025)

[2] Offline Trajectory Optimization for Offline Reinforcement Learning (KDD 2025)

It would greatly strengthen the paper if the authors could explicitly compare BiTrajDiff with these methods. Specifically, what are the theoretical or empirical advantages of the proposed approach over them? For instance, it appears that the method proposed in [1] might inherently possess the capability to resolve the "cross-door" issue as well.

2. Unclear uniqueness of the "cross-door" challenge: The manuscript highlights the "cross-door" issue as a primary challenge; however, it is not entirely clear why this problem is fundamentally more difficult than other trajectory stitching tasks evaluated in the literature, such as the PointMaze environment in [3] (Free from Bellman Completeness: Trajectory Stitching via Model-based Return-conditioned Supervised Learning, ICLR 2024). Why would existing stitching methods, especially model-based approaches that naturally stitch trajectories via rollouts, fail on the cross-door task while succeeding in PointMaze? Clarifying this distinction is crucial for motivating the problem setting.

3. Mechanistic justification for BiTrajDiff: Building on the previous point, the paper would benefit from a deeper explanation of exactly why the BiTrajDiff mechanism succeeds at the cross-door task. Could the authors provide a more rigorous intuition or theoretical insight into which specific property of this bidirectional mechanism overcomes the bottlenecks that cause other algorithms to fail?

4. Algorithmic complexity and compounding biases: The overall framework of BiTrajDiff is quite complex, comprising at least three distinct stages: Generation, Completion, and Filtering. This pipeline raises two primary concerns. First, the computational overhead appears substantial, as it requires Diffusion (Generation), Rollouts (Completion), and Ranking (Filtering). Second, each stage relies on empirical models, which introduces the significant risk of compounding estimation errors and model biases throughout the pipeline. Could the authors comment on the overall computational cost and discuss how they mitigate the accumulation of these biases?

5. Risks associated with the OOD trajectory filter: While the OOD trajectory filter is intended to discover unseen or novel trajectories, this approach seems inherently risky. It introduces the danger of generating completely unrealistic (out-of-distribution) transitions that violate the underlying environmental dynamics. It would be helpful if the authors could provide justification or empirical evidence ensuring that these novel trajectories remain physically plausible and valid within the MDP.

6. Misalignment between core claims and evaluation benchmarks: The central claim of the paper rests on overcoming "hard trajectory stitching" problems. However, the experiments are primarily conducted on standard, widely used benchmarks that may not adequately isolate or stress-test this specific capability. To convincingly validate the method, it would be beneficial to evaluate BiTrajDiff on environments or datasets explicitly designed to require complex trajectory stitching (e.g., highly fragmented datasets with severe spatial bottlenecks).

---

> ### Author Rebuttal · Authors · 2026-03-31
>
> **W1&W6. Need discussion with trajectory stitching methods and more validation of stitching capability**
>
> Thanks for the practical comment. Beyond DiffStitch, prior stitching methods mainly rely on autoregressive rollout with learned world models [1] or direct middle-segment generation between nearby trajectories [2], both of which can make dynamic validity hard to guarantee. In contrast, BiTrajDiff uses shared-anchor bidirectional generation, followed by completion and filtering, to produce novel trajectories with stronger dynamic consistency. Its effectiveness is validated on stitch/explore OGBench and D4RL in Tables R1 and R2, and we will include a fuller comparison in the revision.
>
> Table R1. Offline GCRL comparison on OGBench.
> ||GCIQL(original)|GCIQL(SynthER)|GCIQL(SCoTS)|GCIQL(CompDiffuser)|GCIQL(BiTrajDiff)|CRL(original)|CRL(SynthER)|CRL(SCoTS)|CRL(CompDiffuser)|CRL(BiTrajDiff)|
> |-|-|-|-|-|-|-|-|-|-|-|
> |antmaze-medium-stitch-v0|29±6|31±3|35±2|34±4|**38**±2|53±6|48±3|65±3|61±4|**68**±9|
> |antmaze-large-stitch-v0|7±2|3±4|7±1|7±3|**8**±4|11±2|12±2|**19**±1|17±2|**19**±2|
> |antmaze-medium-explore-v0|13±2|12±3|18±3|17±5|**19**±4|3±2|3±1|**15**±3|11±2|14±2|
> |Avg.|16.3|15.3|20.0|19.3|**21.7**|22.3|21.0|33.0|29.7|**33.7**|
>
> Table R2.  Offline RL comparison on D4RL.
> ||IQL(OTTO)|IQL(BiTrajDiff)|TD3BC(OTTO)|TD3BC(BiTrajDiff)|
> |-|-|-|-|-|
> |halfcheetah-m|**49.3**±0.1|48.6±0.2|49.9±0.2|**50.3**±0.4|
> |hopper-m|78.6±3.5|**81.1**±5.0|74.5±3.7|**79.0**±3.5|
> |walker-m|83.5±1.5|**86.7**±1.7|83.7±0.1|**86.7**±1.0|
> |halfcheetah-m-r|**44.8**±0.1|44.1±0.2|**45.8**±0.2|45.0±0.6|
> |hopper-m-r|102.4±1.0|**102.8**±0.6|80.8±13.2|**85.0**±10.3|
> |walker-m-r|86.9±2.0|**87.6**±2.2|**90.7**±0.3|89.9±1.4|
> |antmaze-umaze-diverse|60.4±2.3|**63.4**±7.9|**53.3**±5.4|47.0±2.1|
> |antmaze-medium-diverse|67.8±2.5|**86.2**±5.8|0.5±0.1|**4.8**±1.7|
> |antmaze-large-diverse|36.8±2.1|**63.0**±9.3|0.0±0.0|**1.2**±0.9|
> |Avg.|67.8|**73.7**|53.2|**54.3**|
>
> **W2. Unclear motivation of the cross-door challenge**
>
> Thanks for the important comment. We do not use the cross-door example to claim a universally harder setting than prior benchmarks like PointMaze. Instead, it is a simple and controlled case for isolating a specific failure mode: although the dataset already covers states near the bottleneck, prior single-direction DA methods still fail to explicitly bridge different behavior patterns. Although rollout-based methods[3] can stitch such patterns, their autoregressive generation may accumulate modeling errors and lead to local inconsistency. Instead, BiTrajDiff produces more stable stitched trajectories by recombining trajectories through anchor-centered bidirectional generation. This is also consistent with our cross-door visualization in this [link](https://anonymous.4open.science/r/r4-00FE/), where MBRCSL often yields invalid behaviors such as wall-crossing.
>
> **W3. Unclear intuition behind the advantage of the bidirectional mechanism**
>
> Thanks for the insightful comment. Prior single-direction DA methods largely preserve the original connectivity of the dataset, which makes recombining different behavior patterns difficult. BiTrajDiff addresses this by decomposing the problem into two simpler and independent diffusion tasks around a shared anchor state: how to arrive and how to continue. This bidirectional design makes novel trajectory synthesis more stable.
>
> **W4&W5. Mitigating bias accumulation and ensuring trajectory plausibility**
>
> Thanks for the important comment. BiTrajDiff mitigates bias accumulation through shared-anchor generation around support-aware regions, OOD filtering to remove overly deviant candidates. Notably, the OOD filter is not designed to seek novelty, but to reject unreliable outliers. As shown in Table R3, the OOD filter consistently lowers dynamic error while the final OOD+Greedy setting still maintains substantial $\mathcal E_{\text{L2D}}$, indicating BiTrajDiff retains novel yet plausible trajectories.
>
> Table R3. Dynamic error and L2 Distance under different filtering strategies.
> |$\mathcal E_\text{Dyn}/\mathcal E_\text{L2D}$|w/o OOD + Greedy|w OOD, w/o Greedy|w Greedy, w/o OOD|w OOD + Greedy Filter(Ours)|
> |-|-|-|-|-|
> |halfcheetah-m-e|0.29/9.39|0.15/8.73|0.36/11.27|0.18/10.45|
> |hopper-m-e|0.44/15.04|0.23/15.61|0.71/18.58|0.30/16.89|
> |walker-m-e|0.34/11.26|0.22/10.36|0.47/14.97|0.26/14.10|
>
> **W4. The overall computational cost of BiTrajDiff**
>
> Thanks for the constructive comment. Due to the lightweight design of the completion and filtering mechanism, the time consumed is in the same range as other DA methods, as shown in Table R4.
>
> Table R4. Time comparison of DA methods.
> ||SynthER|RTDiff|DiffStitch|Ours|
> |-|-|-|-|-|
> |Training|7h|7.5h|11h|9h|
> |Generation|180s|250s|400s|330s|
>
> [1] Offline Trajectory Optimization for Offline Reinforcement Learning
>
> [2] State-covering trajectory stitching for diffusion planners
>
> [3] Free from Bellman Completeness: Trajectory Stitching via Model-based Return-conditioned Supervised Learning

---

> > ### Author Rebuttal · Reviewer_AYYz · 2026-04-01
> >
> > Thanks for the authors detailed feedback! If the following problem can be solved, I will consider raising my score:
> >
> > Regarding Weakness 2, while I appreciate the clarification of your original motivation, I suggest delving deeper into the "cross-door" problem. It presents an intriguing phenomenon: specifically, why do existing methods like single-direction DA fail in this scenario? Why this kind of benchmarks more difficult than others, better from mathematical or theoretical view?
> >
> > Identifying the underlying reasons for this failure would significantly strengthen the theoretical foundation of your work, making it more clearly distinguishing it from prior research. Thanks!

---

> > > ### Author Response · Authors · 2026-04-02
> > >
> > > Thanks for this helpful follow-up. We would like to clarify that the cross-door example is a simple and controlled setting designed to isolate a specific failure mode: the dataset may provide sufficient local support around an anchor state, while still lacking the trajectory-level connectivity needed to bridge different behavior patterns. (e.g., incoming and outgoing local segments are both observed near the same anchor, but their cross-pattern pairing is not realized in the dataset).
> > >
> > > More formally, let $\mathcal D$ denote the offline dataset, where each $\tau\in\mathcal D$ is a trajectory. For a given anchor state $s$, let $h^{-}$ denote a local history segment arriving at $s$, and $h^{+}$ denote a local future segment departing from $s$. We denote by
> > > $$
> > > \mathcal H_{\mathcal D}(s)=\\{h^-:\exists \tau\in\mathcal D,\ (h^-,s)\subset\tau\\}
> > > $$
> > > the set of incoming local patterns observed around $s$, and by
> > > $$
> > > \mathcal F_{\mathcal D}(s)=\\{h^+:\exists \tau\in\mathcal D,\ (s,h^+)\subset\tau\\}
> > > $$
> > > the set of outgoing local patterns. The actually observed couplings between them are
> > > $$
> > > \mathcal C_{\mathcal D}(s)=\\{(h^-,h^+):\exists \tau\in\mathcal D,\ (h^-,s,h^+)\subset\tau\\},
> > > $$
> > > which is generally a sparse subset of $\mathcal H_{\mathcal D}(s)\times \mathcal F_{\mathcal D}(s)$.
> > >
> > > More precisely, the cross-door task isolates the case **where doorway-adjacent states $s_d$ have sufficient one-sided support, but the observed coupling set $\mathcal C_{\mathcal D}(s_d)\subseteq \mathcal H_{\mathcal D}(s_d)\times \mathcal F_{\mathcal D}(s_d)$ occupies only a small portion of the full pairing space. Thus, what is missing is not local state support, but support in the joint connectivity structure between incoming and outgoing trajectory patterns.** Meanwhile, we do not view this benchmark as universally harder than others, but as more diagnostic for our purpose, because it suppresses the easier explanation of missing local support and **makes success depend more directly on recovering unseen pairings.**
> > >
> > > This also explains why single-direction DA tends to fail in this setting. Such methods are trained to model one-directional continuations sampled from the dataset, so **their supervision is concentrated on the observed couplings in $\mathcal C_{\mathcal D}(s)$**. If a cross-pattern pairing never appears in $\mathcal D$, it receives little or no direct training signal, and the learned generator tends to preserve the original connectivity structure rather than create new pairings. By contrast, BiTrajDiff factorizes local bridge generation around the same anchor into two conditionals, one for incoming segments and one for outgoing segments, and then recombines them. As a result, **it is no longer tied only to the observed coupling set $\mathcal C_{\mathcal D}(s)$, but can represent candidate pairings from a much richer space induced by $\mathcal H_{\mathcal D}(s)\times \mathcal F_{\mathcal D}(s)$.** Completion and filtering then retain only the feasible subset. We believe this is the key theoretical intuition for why the bidirectional design is effective in the cross-door setting.
> > >
> > >  We hope this response addresses your concerns.

---

### Official Review · Reviewer_cCds · 2026-03-13

**Soundness:** 3
**Presentation:** 4
**Significance:** 2
**Originality:** 2
**Overall Recommendation:** 4
**Confidence:** 4

**Summary:**

This paper proposes BiTrajDiff, a bidirectional diffusion framework that synthesizes both forward-future and backward-history trajectory segments from shared anchor states to recover global connectivity in offline datasets. By bridging disconnected behavioral modes, it overcomes the limitations of single-directional rollout methods that often preserve dataset biases and lack diversity.

**Compliance With Llm Reviewing Policy:**

Affirmed.

**Final Justification:**

Since the rebuttal has mainly addressed my questions. I don't have further concerns about this paper.

**Key Questions For Authors:**

See weakness above.

**Limitations:**

Yes

**Strengths And Weaknesses:**

**Strength**:
1. This paper introduced an interesting limitation of existing data augmentation methods in offline reinforcement learning. The authors used a clear toy example to show the effectiveness and importance of BiTrajDiff.
2. The experiment results show that BiTrajDiff got a strong performance in D4RL, achieved the SOTA performance.

**Weakness**:

1. The paper is quite similar with previous work GTA[1]. The authors didn't compare with GTA either in related works and in the experiments.

2. BiTrajDiff is complicated with multiple components. The authors need to discuss about the effectiveness of BiTrajDiff and provide a detailed running time.

3. In the experiments, the authors set the generation length $H=5$, which sounds like a very short generation length. Have the authors tried different length setting and explain why the generation length is set to be so small? Such a short generation length cannot bring too much information as stated in the toy example.

4. How did the authors choose the anchor states? How to ensure that those anchor states can bring faithful information beyond the offline dataset.


[1] Lee, Jaewoo, et al. "Gta: Generative trajectory augmentation with guidance for offline reinforcement learning." Advances in Neural Information Processing Systems 37 (2024): 56766-56801.

---

> ### Author Rebuttal · Authors · 2026-03-31
>
> **W1. Compare with GTA [1] and discuss with it**
>
> Thanks for the practical suggestion. GTA augments offline datasets by partially noising existing trajectories and denoising them through amplified reward guidance to refine behavioral quality. However, GTA still follows the single-direction paradigm, mainly refining existing trajectory structure rather than creating new connectivity between disconnected behavior patterns. By contrast, BiTrajDiff uses **shared-anchor bidirectional generation** to explicitly recombine different patterns. The results in Table R1 support the superiority of BiTrajDiff.
>
> Table R1. Comparison of GTA and BiTrajDiff in navigation tasks.
> ||IQL(GTA)|IQL(BiTrajDIff)|TD3BC(GTA)|TD3BC(BiTrajDiff)|
> |-|-|-|-|-|
> |maze2d-umaze|41.7±1.4|**62.3**±2.7|**46.1**±10.5|45.3±3.1|
> |maze2d-medium|37.8±1.7|**72.2**±14.8|42.6±8.7|**45.7**±7.6|
> |maze2d-large|**76.7**±4.7|71.3±2.3|112.7±30.9|**129.3**±26.0|
> |antmaze-umaze-diverse|57.9±9.5|**63.4**±7.9|44.6±9.3|**47.0**±2.1|
> |antmaze-meidum-diverse|78.1±7.8|**86.2**±5.8|0.0±0.0|**4.8**±1.7|
> |antmaze-large-diverse|47.8±6.7|**63.0**±9.3|0.0±0.0|**1.2**±0.9|
> |kitchen-complete|57.9±9.5|**59.1**±12.3|0.6±0.2|**1.5**±1.0|
> |kitchen-partial|45.9±6.4|**69.7**±1.8|**15.8**±12.4|13.0±5.9|
> |kitchen-mixed|56.2±1.9| **62.9**±4.9|21.0±4.3|**30.9**±12.6|
> |Avg.|55.6|**67.8**|31.5|**35.5**|
>
>
> **W2. Discuss about the effectiveness of BiTrajDiff and provide a detailed running time.**
>
> Thanks for this insightful comment. The practical cost remains moderate: training only requires a single 80GB A100 GPU, and the lightweight completion/filtering stages keep the overall runtime in the same range as other DA baselines, as shown in Table R2.
>
> Table R2. Training and generation time comparison of DA methods.
> ||SynthER|RTDiff|DiffStitch|Ours|
> |-|-|-|-|-|
> |Training|7h|7.5h|11h|9h|
> |Generation|180s|250s|400s|330s|
>
>
> **W3. Sensitivity and rationale of the generation length setting**
>
> Thanks for this important comment. We would like to clarify that the horizon $H$ in BiTrajDiff refers to the length generated by each single diffusion model, not the final stitched trajectory. After bidirectional stitching, the effective bridge span is $2H-2$ transitions; thus, with the default $H=5$, the final stitched trajectory already covers an 8-step span around the anchor, which is not unusually short. This scale is also comparable to prior trajectory-level DA methods, e.g., RTDiff uses a similar default horizon ($H=10$).
>
> We further performed a horizon ablation in Table R3, and the results are consistent with the observation also reported in RTDiff when the generated trajectory becomes too long, performance starts to decline. Intuitively, longer generated segments are harder to model accurately, and the resulting modeling error can further propagate through downstream Offline RL TD estimation, eventually affecting offline policy learning. Therefore, a moderate horizon provides the best trade-off between additional information and data reliability.
>
> Table R3. Horizon ablation on walker-medium-replay-v2.
>
> |Horizon $H$|Effective bridge length $2H-2$|IQL| TD3BC|
> |-|-|-|-|
> |3|4|6.4±6.3|3.9±4.6|
> |5|8|**14.3**±2.2|**9.0**±1.4|
> |8|14|13.9±2.7|8.2±3.9|
> |10|18|9.8±4.5|7.5±4.2|
> |20|38|-0.5±7.4|-2.8±10.7|
>
> **W4. Anchor state selection and faithfulness**
>
> Thanks for the insightful question. In our implementation, anchor states are randomly sampled from the offline dataset, which provides broad coverage over the dataset support. Their role is not to create far-OOD states lacking true environment dynamics supervision, but to act as shared conditioning points for bridging distinct behavior patterns and recovering trajectory-level connections absent from the original dataset. Faithfulness is ensured by both generation and filtering. The directional diffusion models are trained on the offline data distribution, and the filtering stage further removes implausible candidates: Table R4 shows that OOD filtering significantly reduces $\mathcal E_{\text{Dyn}}$, while the final OOD+Greedy setting preserves substantial $\mathcal E_{\text{L2D}}$. This indicates that BiTrajDiff does not rely on anchors to invent arbitrary new states, but to produce faithful new connectivity patterns around states already supported by the dataset.
>
> Table R4. Dynamic error and L2 Distance under different filtering strategies.   $ \mathcal E_\text{Dyn}(\hat \tau)=\sum_{t}\\lVert\hat s_{t+1}-s_{t+1}\\rVert_2$
> , where $s_{t+1}$ is computed from the true dynamics $f(\hat s_t,\hat a_t)$;  $\mathcal E_\text{L2D}(\hat{\tau})=\min_{\tau\in\mathcal D}\sum_t ||\hat{s}_t-s_t||_2$, where $s_t\in\tau$.
> |$\mathcal E_\text{Dyn}/\mathcal E_\text{L2D}$|w/o OOD + Greedy|w OOD, w/o Greedy|w Greedy, w/o OOD|w OOD + Greedy Filter (Ours)|
> |-|-|-|-|-|
> |halfcheetah-m-e|0.29/9.39|0.15/8.73|0.36/11.27|0.18/10.45|
> |hopper-m-e|0.44/15.04|0.23/15.61|0.71/18.58|0.30/16.89|
> |walker-m-e|0.34/11.26|0.22/10.36|0.47/14.97|0.26/14.10|

---

> > ### Author Rebuttal · Reviewer_cCds · 2026-04-01
> >
> > Thanks for the reply. My most concerns are resolved.

---

> > > ### Author Response · Authors · 2026-04-02
> > >
> > > Thank you for your feedback and for updating your score. We’re glad our responses addressed your concerns.

---

### Official Review · Reviewer_mqff · 2026-03-13

**Soundness:** 3
**Presentation:** 2
**Significance:** 3
**Originality:** 2
**Overall Recommendation:** 4
**Confidence:** 3

**Summary:**

In this paper, the authors study the limitations of regular offline reinforcement learning (RL) data augmentation (DA) methods in generating data with novel trajectory-level connectivity. The authors propose Bidirectional Trajectory Diffusion (BiTrajDiff), a data augmentation framework that synthesize novel trajectories by conditioning on shared anchor states to generate coherent forward-future and backward-history segments. The generated trajectories are further filtered with an OOD filter and a return filter, before being used for policy learning. The proposed method is evaluated on offline RL benchmarks (D4RL), showing improvements over other augmentation baselines and generalizability for different offline RL methods.

**Compliance With Llm Reviewing Policy:**

Affirmed.

**Final Justification:**

The paper presents a clean and well-motivated framework for recovering trajectory-level connectivity in offline RL through bidirectional diffusion generation. The strengths lie in clear motivation, systematic ablations, and consistent improvements across multiple offline RL backbones.

My initial concerns centered on insufficient discussion of related stitching methods, lack of failure case analysis, missing evaluations on modern benchmarks like OGBench, and unclear differentiation from DiffStitch. The authors' rebuttal adequately addressed all of these: they provided OGBench comparisons showing competitive performance against recent stitching baselines (SCoTS, CompDiffuser), offered quantitative trajectory-level analysis (dynamic error and L2 distance) clarifying how BiTrajDiff differs from DiffStitch, included informative filtering ablations, and discussed failure cases tied to backbone limitations. I am satisfied with the responses and have raised my score to 4 (weak accept) accordingly.

**Key Questions For Authors:**

1. Could authors add a more detailed comparison between BiTrajDiff and DiffStitch? The differences between them are not sufficiently discussed. Is there an analysis comparing the types of trajectories each method generates?
2. Could authors add a discussion about relevant diffusion stitching work?
3. What fraction of generated trajectories filtered by the two filters? Does the filter (especially OOD filter) remove trajectories that have novel and feasible connectivity patterns that we want?
4. If feasible, could the authors add comparisons on modern benchmarks (e.g., OGBench) and compare with more recent trajectory stitching methods?

**Limitations:**

The paper discusses some technical limitations, but the limitation discussion could be more explicit about dependence on accurate inverse dynamics and reward models, sensitivity and reliance on filtering/selection mechanism. I do not see potential societal risks beyond the standard caution.

**Strengths And Weaknesses:**

**Strengths**

* The paper conveys a clear and intuitive motivation. I generally like the two-room toy task example, which effectively communicates the paper's central focus.

* The overall framework is conceptually clean. The introduction of bidirectional generation enriches more diverse data and the filtering is important for stabilizing synthetic data augmentation.

* The authors present thorough experimental analysis and ablation studies. The framework's effectiveness and outcomes of design choices are systematically validated across different scenarios (dense reward, sparse reward) and different offline RL methods (IQL/TD3BC/DT).

**Weaknesses**

* This paper should discuss the line of trajectory stitching as relevant work, e.g. [1][2]. They are not directly applied to data augmentation but are highly relevant in the methodology level. If feasible, the experimental comparisons can also be improved by adding them as baselines.

* The paper does not analyze failure cases, where the proposed framework leads to modest or no improvement (e.g., antmaze-medium/large-diverse). The authors should add failure case analysis to provide more insights into the  method's applicability and limitations.

* The D4RL benchmark is standard but outdated for evaluating trajectory stitching. The experiments could be improved by adding evaluations on OGBench, which is specifically designed for testing stitching and bridging local patterns.

[1] Luo, Yunhao, et al. "Generative trajectory stitching through diffusion composition." arXiv preprint arXiv:2503.05153 (2025).

[2] Lee, Kyowoon, and Jaesik Choi. "State-covering trajectory stitching for diffusion planners." arXiv preprint arXiv:2506.00895 (2025).

---

> ### Author Rebuttal · Authors · 2026-03-31
>
> **W1&W3&Q2&Q4. Need discussion and comparison with trajectory stitching methods in OGBench**
>
> Thanks for the practical suggestion. Besides DiffStitch, trajectory stitching is also achieved by composing short trajectory chunks into longer goal-conditioned trajectories through compositional sampling [1], or direct middle-segment generation under the temporal distance-preserving latent space [2]. Rather than relying on autoregressive rollout or entire middle segment generation which is hard to guarantee the dynamics validity, BiTrajDiff recombines behavior patterns through shared-anchor bidirectional generation, and then validates through completion and filtering to obtain novel trajectories with strong dynamic consistency. The effectiveness of BiTrajDiff against these methods is validated on OGBench in Table R1. We will also include more complete experiments and discussion in the revision.
>
> Table R1. Offline GCRL comparison on OGBench.
> |Env.|GCIQL(original)|GCIQL(SynthER)|GCIQL(SCoTS)|GCIQL(CompDiffuser)|GCIQL(BiTrajDiff)|CRL(original)|CRL(SynthER)|CRL(SCoTS)|CRL(CompDiffuser)|CRL(BiTrajDiff)|
> |-|-|-|-|-|-|-|-|-|-|-|
> |antmaze-medium-stitch-v0|29±6|31±3|35±2|34±4|**38**±2|53±6|48±3|65±3|61±4|**68**±9|
> |antmaze-large-stitch-v0|7±2|3±4|7±1|7±3|**8**±4|11±2|12±2|**19**±1|17±2|**19**±2|
> |antmaze-medium-explore-v0|13±2|12±3|18±3|17±5|**19**±4|3±2|3±1|**15**±3|11±2|14±2|
> |Avg.|16.3|15.3|20.0|19.3|**21.7**|22.3|21.0|33.0|29.7|**33.7**|
>
> **W2. Failure case analysis**
>
> We appreciate the suggestive comment. An important observation is that, across all settings we considered, BiTrajDiff consistently improves performance over other DA methods, suggesting that the augmented data are broadly useful. However, the final performance still depends strongly on the offline RL backbone. In challenging settings such as *antmaze-medium/large*, relying heavily on accurate long-horizon value estimation, the policy-constrained method TD3+BC and the return-conditioned supervised method DT tend to significantly underperform the value-regularized methods IQL and CQL. Thus, the overall gain of BiTrajDiff is naturally limited. A further limitation of BiTrajDiff is the reliance on learned inverse dynamics and reward models, whose labeling error may increase in higher-dimensional settings and limit downstream gains. This is an important direction for future work.
>
> **Q1.Detailed comparison between BiTrajDiff and DiffStitch, as well as other DA methods.**
>
> Thank you for the insightful comment. We quantitatively compare the dynamic error $\mathcal E_\text{Dyn}(\hat \tau)$ and L2 Distance $\mathcal E_\text{L2D}(\hat \tau)$ (described in Section 5.5 of the main text) in Tables R2 to highlight the difference between the DA methods. Meanwhile, we also conducted a toy example visualization in this [link](https://anonymous.4open.science/r/work-767C). SynthER-long and RTDiff achieve relatively low dynamic error, but show limited ability to form new trajectory-level connections. DiffStitch increases diversity, but at the cost of substantially worse dynamic consistency. In contrast, BiTrajDiff generates novel trajectories from a shared anchor in both directions, which better preserves local dynamics while recovering new connectivity patterns.
>
> Table R2. Dynamic error and L2 Distance of different DA methods.
> |$\mathcal E_\text{Dyn}/\mathcal E_\text{L2D}$|SynthER-long|RTDiff|DiffStitch|BiTrajDiff (Ours)|
> |-|-|-|-|-|
> |halfcheetah-m-e|0.12/6.51|0.14/7.30|0.18/10.24|0.18/10.45|
> |hopper-m-e|0.22/12.36|0.31/13.97|0.42/19.23|0.30/16.89|
> |walker-m-e|0.21/9.07|0.26/11.41|0.35/14.82|0.26/14.10|
>
> **Q3. What fraction of the generated trajectories filtered by the two filters? Does the filter remove novel trajectories?**
>
> We appreciate the insightful question. As described in Appendix B.2.1, from 512 generated trajectories, the OOD filter retains 256, and the greedy filter further keeps 64, so only 12.5% are finally retained. Meanwhile, the OOD filter is not designed to remove novelty, but to remove unreliable novelty, while the greedy filter keeps more useful trajectories among the remaining candidates.
> To clarify the effect of the two filters, we report dynamic error and L2 Distance in Table R3. The OOD filtering significantly reduces $\mathcal E_{\text{Dyn}}$, while the final OOD+Greedy setting keeps low dynamic error without collapsing the L2 Distance, indicating high reliability while preserving the novelty introduced by bidirectional generation.
>
> Table R3. Dynamic error and L2 Distance under different filtering strategies.
> |$\mathcal E_\text{Dyn}/\mathcal E_\text{L2D}$|w/o OOD + Greedy|w OOD, w/o Greedy|w Greedy, w/o OOD|w OOD + Greedy Filter (Ours)|
> |-|-|-|-|-|
> |halfcheetah-m-e|0.29/9.39|0.15/8.73|0.36/11.27|0.18/10.45|
> |hopper-m-e|0.44/15.04|0.23/15.61|0.71/18.58|0.30/16.89|
> |walker-m-e|0.34/11.26|0.22/10.36|0.47/14.97|0.26/14.10|
>
> [1] Generative trajectory stitching through diffusion composition
>
> [2] State-covering trajectory stitching for diffusion planners

---

> > ### Author Rebuttal · Reviewer_mqff · 2026-04-04
> >
> > I thank the authors for the detailed and clear rebuttal. The clarifications and additional experiments have largely addressed my concerns, and I plan to raise my score to 4 accordingly. I look forward to reading the full additional analysis and discussion in the revised paper.

---

> > > ### Author Response · Authors · 2026-04-04
> > >
> > > Thank you for your feedback and for updating your score. We sincerely appreciate your time and careful evaluation of our manuscript.

---

### Official Review · Reviewer_e5mg · 2026-03-13

**Soundness:** 3
**Presentation:** 3
**Significance:** 3
**Originality:** 2
**Overall Recommendation:** 3
**Confidence:** 4

**Summary:**

This paper proposes **BiTrajDiff**, a data augmentation method for offline RL aimed at improving trajectory-level connectivity in offline datasets. The method generates forward and backward trajectory segments around a shared anchor state, stitches them into new trajectories, and then completes and filters the generated samples for downstream training. Experiments on D4RL with several offline RL algorithms show gains over prior augmentation methods, including Synther, DiffStitch, and RTDiff.

**Compliance With Llm Reviewing Policy:**

Affirmed.

**Final Justification:**

Final Justification

The paper presents a bidirectional trajectory generation framework and is generally technically sound. The rebuttal adds useful empirical evidence (runtime, dynamics, OGBench), which helps clarify performance.

However, my main concern about the necessity and distinct role of the bidirectional design remains only partially addressed. While the connectivity perspective is better articulated, it is still unclear whether the gains specifically rely on this design, or could be achieved through alternative formulations of comparable complexity. In addition, the degree of conceptual novelty beyond existing stitching or conditional generation approaches appears somewhat incremental.

Overall, the work is promising, but the core design choice is not yet fully justified. The rebuttal improves clarity but does not substantially change my assessment, so I maintain a borderline reject recommendation.

**Key Questions For Authors:**

1. The bidirectional forward/backward generation design is the central contribution of the paper. Can the authors clarify what specific capability this design provides beyond single-direction generation, and why this capability requires modeling both directions explicitly rather than using a simpler alternative formulation?

2. Since the anchor states are always sampled from the offline dataset, and the generated trajectories are further constrained by OOD filtering, can the authors clarify in what sense BiTrajDiff expands trajectory connectivity beyond the original dataset support, rather than mainly recombining patterns around observed states?

3. The core operation of the method is stitching two generated segments around the anchor. Can the authors provide more direct evidence that the stitched transitions near the anchor are locally feasible and dynamically consistent, beyond downstream return improvements and coarse global trajectory metrics?

4. The paper’s central claim is that BiTrajDiff recovers missing trajectory-level connectivity, but the evaluation is limited to standard D4RL settings. These benchmarks are not sufficient to directly validate that claim, since performance gains there may also arise from generic data augmentation effects. The authors should evaluate the method on more challenging benchmarks explicitly designed to test stitching, long-horizon composition, or connectivity recovery, such as **OGBench**-style tasks. Without such evidence, it is difficult to determine whether the method truly addresses the claimed problem.

5. The paper is closely related to recent stitching-oriented literature, including methods that explicitly discuss feasibility, reachability, and support expansion. Can the authors clarify how BiTrajDiff should be positioned relative to closely related work such as ASTRO and SSD, both conceptually and in terms of claimed novelty?

**Limitations:**

The paper does not sufficiently discuss several important limitations, including the added complexity of the bidirectional design, the possibility that anchor-conditioned generation remains close to the original data support, the lack of direct validation of stitching feasibility near the anchor, and the limited evaluation setting for the paper’s main connectivity-recovery claim.

**Strengths And Weaknesses:**

**Strengths**

1. **Clear motivation.** The paper identifies a meaningful limitation of prior augmentation methods: single-direction generation tends to preserve the dataset’s original trajectory connectivity and therefore cannot effectively bridge disconnected behavior modes. This is a good framing, and the toy example makes the intuition clear.

2. **Simple and intuitive high-level idea.** The shared-anchor bidirectional generation mechanism is easy to understand and aligned with the stated motivation. Using a common anchor to connect generated past and future segments is a natural design choice.

3. **Reasonable experimental coverage within D4RL.** The paper compares against strong augmentation baselines and includes ablations on forward-only, backward-only, filtering, and related design choices. The bidirectional-vs-single-direction ablation is particularly important and supports the main design intuition.

**Weaknesses**

1. **The necessity of the bidirectional design is not fully established.**
   The bidirectional forward/backward generation design is the central contribution of the paper, and the experiments do suggest that it is useful. However, the paper does not clearly isolate what specific capability is gained by explicitly modeling both directions, beyond showing that the full method performs better than single-direction variants. As a result, it remains unclear why this design should be viewed as the most appropriate formulation of the bridge-generation problem, rather than one possible construction among several alternatives.

2. **The additional modeling and computational complexity is not sufficiently justified.**
   The full method requires two diffusion models, trajectory completion, and filtering. This makes the overall pipeline heavier than a simpler alternative might be. However, the paper does not clearly explain whether the empirical improvement is substantial enough to justify this added complexity, nor does it provide a detailed discussion of the associated training and sampling overhead.

3. **Because the anchor is sampled from the offline dataset, the method may still remain confined to local recomposition rather than truly overcoming the dataset support limitation.**
   This is a central concern for a data augmentation method. Although the paper argues that bidirectional stitching can recover missing trajectory connectivity, both the forward and backward generations are conditioned on observed in-dataset anchor states, and the final OOD filter further suppresses trajectories that deviate too far from the original data distribution. As a result, it is unclear whether BiTrajDiff truly expands the effective support of the dataset, or mainly performs a stronger form of local recomposition around existing states. This matters because overcoming dataset support limitations is precisely the core promise of augmentation in offline RL.

4. **The paper does not directly analyze stitching validity near the anchor region.**
   Since the key operation is stitching independently generated backward-history and forward-future segments around the same anchor, the most important question is whether the local transitions near the stitching point are actually feasible and dynamically consistent. The current evidence mainly relies on downstream RL performance and coarse global trajectory metrics, which is not sufficient to validate the core stitching mechanism itself.

5. **There is a mismatch between the paper’s claim and its evaluation.**
   The paper is motivated as recovering missing **trajectory connectivity**, but the experiments are entirely on **D4RL**, with much of the evidence coming from standard MuJoCo locomotion tasks. These environments are useful, but they are not the most direct benchmarks for testing stitching or compositional connectivity recovery. As a result, it remains unclear whether the reported gains really come from recovering disconnected behavior modes, or simply from generic data augmentation effects. A stronger evaluation should include benchmarks explicitly designed for stitching or compositional generalization, such as **OGBench**-style tasks.

6. **The related-work discussion is incomplete and should better position the paper against nearby stitching-based methods.**
   The paper does compare against RTDiff and DiffStitch, so the baseline coverage is not poor. However, for a paper centered on trajectory stitching and connectivity recovery, the literature positioning still feels incomplete. In particular, **ASTRO** is highly relevant because it also studies stitching under dynamics and reachability constraints, and is closely related in spirit to the paper’s central motivation. Even if the authors do not include it as an empirical baseline, I believe it should at least be discussed in the related work or positioning section. More broadly, the paper would also benefit from clearer positioning relative to diffusion-based stitching works such as **SSD** and other recent stitching-oriented approaches. [1][2]

[1] SSD: diffusion-based sub-trajectory stitching for offline goal-conditioned RL.
[2] ASTRO: stitching-based offline RL with emphasis on reachability and dynamics consistency.

---

> ### Author Rebuttal · Authors · 2026-03-31
>
> **W1&W4&Q1&Q3. Why is bidirectional generation necessary, and what kind of ''support expansion'' does BiTrajDiff actually provide?**
>
> Thanks for these important comments. We would like to clarify that the significant role of our bidirectional design is **anchor-centered trajectory connectivity recovery that prior DA methods largely overlook**. Bridge generation around an anchor requires modeling both plausible histories that arrive at the anchor and plausible futures that depart from it. The bottleneck-adjacent states may already exist in the dataset, while the predecessor–successor combinations required to connect them are absent. While single-direction DA methods capture only one side of this structure tend to preserve the original connectivity and mode composition of the dataset, BiTrajDiff explicitly models both sides around the same anchor and recombines them to recover missing trajectory-level combinations over support-aware regions. Meanwhile, the cross-door task in Fig. 1 of the main text makes this distinction clear: BiTrajDiff can generate novel cross-door trajectories while single-direction methods mainly model within-room variation and rarely bridge the doorway.  The same interpretation is consistent with our trajectory-distance analysis in Fig. 7 of the main text: single-direction generated trajectories remain close to the original dataset in trajectory space, indicating limited ability to form new trajectory-level combinations.
>
>
> **W2. The additional modeling and computational complexity is not sufficiently justified.**
>
> Thank you for this insightful comment. We have additionally provided the time consumed comparison in Table R1. The results show that the lightweight completion/filtering stages keep the overall runtime in the same range as strong DA baselines such as RTDiff and SynthER.
>
> Table R1. Training and generation time comparison of DA methods.
> ||SynthER|RTDiff|DiffStitch|Ours|
> |-|-|-|-|-|
> |Training|7h|7.5h|11h|9h|
> |Generation|180s|250s| 400s|330s|
>
>
> **W4&Q3. The paper does not directly analyze stitching validity near the anchor region.**
>
> Thanks for this important comment. BiTrajDiff does not stitch arbitrary segments without constraint: both directional diffusion models are trained on the offline data distribution, and the OOD filter further removes trajectories that deviate strongly from the dataset. Additionally, we measure the one-step dynamics error immediately before and after the anchor: $\epsilon^-=\mathrm{MSE}\left(f(\hat s_{t-1},\hat a_{t-1}),\, s_t\right),\epsilon^+=\mathrm{MSE}\left(f(s_t,\hat a_t),\, \hat s_{t+1}\right),$ where $f$ is the true environment dynamics in Table R2, the resulting errors remain small and comparable, indicating that the stitched transitions are locally feasible rather than introducing implausible transitions. This is also consistent with the toy-task visualization in Fig. 1 of the main text, where BiTrajDiff forms new connectivity patterns without obviously invalid behaviors such as wall-crossing.
>
> Table R2. Anchor-local dynamic error on *halfcheetah-medium-expert-v2*.
> |Algo.|$\epsilon^-  ~(\times 10^{-2})$|$\epsilon^+~(\times 10^{-2})$|
> |-|-|-|
> |SynthER|-|0.8|
> |RTDiff|1.5|-|
> |BiTrajDiff|1.1|1.0|
>
> **W5&W6&Q4&Q5. The paper needs related work discussion with stitching-based methods and comparison on more benchmarks for stitching, like OGBench.**
>
> Thanks for the important comments. Existing stitching methods roughly follow two paradigms: autoregressive rollouts accumulate modeling error along the generated segment[1], and direct middle-segment generation [2]. Their dynamic validity is harder to guarantee. In contrast, BiTrajDiff recombines behavior patterns through bidirectional generation around reliable anchor states. As a result, it generates novel trajectories while maintaining stronger dynamic consistency. The effectiveness of BiTrajDiff against these methods is validated on OGBench in Table R3. We will also include a more complete set of experiments and a clearer discussion in the revised version.
>
> Table R3. Offline RL Performance comparison on selected OGBench.
> |Env.|IQL(original)|IQL(ASTRO)|IQL(DiffStitch)|IQL(SynthER)|IQL(SSD)|IQL(BiTrajDiff)|FQL(original)|FQL(ASTRO)|FQL(DiffStitch)|FQL(SynthER)|FQL(SSD)|FQL(BiTrajDiff)|
> |-|-|-|-|-|-|-|-|-|-|-|-|-|
> |antmaze-large-stitch-singletask-task1|26.2|51.7|35.0|31.1|53±7|**64**±10|29.2|57.3|33.1|28.7|75±22|**79**±13|
> |humanoidmaze-medium-stitch-singletask-task1|29.7|31.4|28.3|31.2|33±11|**38**±7|17.5|30.0|22.6|15.9|43±16|**47**±18|
> |cube-single-play-singletask-task1|81.5| **89.2**|79.0|82.4|82±7|87±5|88.0|92.9|89.6|87.3|93±4|**98**±4|
> |Avg.|45.8|57.4|47.4|48.2|56.0|**63.0**|44.9|60.1|48.4|44.0|70.3|**74.7**|
>
> [1] Stitching Sub-Trajectories with Conditional Diffusion Model for Goal-Conditioned Offline RL
>
> [2] ASTRO: Adaptive Stitching via Dynamics-Guided Trajectory Rollouts

---

> > ### Author Rebuttal · Reviewer_e5mg · 2026-04-04
> >
> > Thank you for the rebuttal. The additional runtime comparison, local dynamics analysis, and OGBench results are helpful. I still have some reservations about the role and necessity of the bidirectional design in the overall method.

---

> > > ### Author Response · Authors · 2026-04-04
> > >
> > > Thank you very much for this helpful follow-up.
> > >
> > > Our main point is that BiTrajDiff is designed to address a **trajectory-level connectivity problem that prior DA methods do not model explicitly**. Specifically, the dataset may already provide sufficient local support around an anchor state, while still lacking the trajectory-level connectivity needed to bridge different behavior patterns.
> > >
> > > More formally, let $\mathcal D$ denote the offline dataset, and for a given anchor state $s$, let $h^{-}$ denote a local history segment arriving at $s$, and $h^{+}$ denote a local future segment departing from $s$. We denote by $\mathcal H_{\mathcal D}(s)=\\{h^-:\exists \tau\in\mathcal D,\ (h^-,s)\subset\tau\\}$ the set of incoming local patterns observed around $s$, and by $\mathcal F_{\mathcal D}(s)=\\{h^+:\exists \tau\in\mathcal D,\ (s,h^+)\subset\tau\\}$ the set of outgoing local patterns. The actually observed couplings between them are $\mathcal C_{\mathcal D}(s)=\\{(h^-,h^+):\exists \tau\in\mathcal D,\ (h^-,s,h^+)\subset\tau\\}$, which is typically a sparse subset of $\mathcal H_{\mathcal D}(s)\times \mathcal F_{\mathcal D}(s)$. Thus, **what is missing is not local state support, but support in the joint connectivity structure between incoming and outgoing trajectory patterns**. This also explains why single-direction DA tends to fail in this setting: since it only models one-directional continuations sampled from the dataset, its supervision is concentrated on the observed couplings in $\mathcal C_{\mathcal D}(s)$, so the learned generator tends to preserve the original connectivity structure rather than create new pairings. By contrast, BiTrajDiff factorizes local bridge generation around the same anchor into two conditionals, one for incoming segments and one for outgoing segments, and then recombines them. **As a result, it is no longer tied only to the observed coupling set $\mathcal C_{\mathcal D}(s)$, but can represent candidate pairings from a much richer space induced by $\mathcal H_{\mathcal D}(s)\times \mathcal F_{\mathcal D}(s)$. We believe this is the key theoretical intuition for why the bidirectional design is effective and necessary for trajectory-level connectivity recovery.**
> > >
> > > Empirically, we also support this explanation from three aspects.
> > >
> > > - we quantitatively compare the dynamic error $\mathcal E_{\text{Dyn}}(\hat\tau)$ and L2 distance $\mathcal E_{\text{L2D}}(\hat\tau)$ of different DA methods below. The results show that single-direction methods such as SynthER-long and RTDiff achieve relatively low dynamic error, but remain limited in forming new trajectory-level combinations. In contrast, **BiTrajDiff achieves a better balance, generating more novel trajectories while maintaining comparable dynamic consistency.**
> > >
> > > | $\mathcal E_\text{Dyn}/\mathcal E_\text{L2D}$ | Synther-long | RTDiff     | BiTrajDiff (Ours) |
> > > | --------------------------------------------- | ------------ | ---------- | ----------------- |
> > > | halfcheetah-m-e                               | 0.12/6.51    | 0.14/7.30  | 0.18/10.45        |
> > > | hopper-m-e                                    | 0.22/12.36   | 0.31/13.97 | 0.30/16.89        |
> > > | walker-m-e                                    | 0.21/9.07    | 0.26/11.41 | 0.26/14.10        |
> > >
> > > - We visualize the generated trajectories of different DA methods in the cross-door task in this [link](https://anonymous.4open.science/r/work-767C/README.md), where the doorway-adjacent states directly instantiate the sparse-coupling case described above. The visualization shows that single-direction DA methods mainly preserve within-room patterns, corresponding to pairings already concentrated around $\mathcal C_{\mathcal D}(s)$, while **BiTrajDiff can recover new cross-door connections without obvious invalid behaviors, corresponding to feasible pairings induced by $\mathcal H_{\mathcal D}(s)\times \mathcal F_{\mathcal D}(s)$ but absent from $\mathcal C_{\mathcal D}(s)$.**
> > >
> > > - We further evaluate on OGBench (Table R3 in the first round rebuttal), which is more directly aligned with stitching-oriented generalization. The gains there suggest that the **bidirectional synthesis mechanism is not only a toy-example effect, but also translates to challenging stitching-style benchmarks**.
> > >
> > > We will incorporate these additional analyses, experiments, and discussions into the revision. Thank you again for your valuable feedback. We hope this addresses your concern.

---

### Decision · Program_Chairs · 2026-04-30

**Decision:**

Accept (regular)

**Comment:**

All reviewers agree that the paper studies a relevant and clearly motivated problem, the proposed framework is intuitive and sound, and the D4RL experiments are convincing. Their main concerns were about a lack of motivation behind the bidirectional design, missing comparison with many related works (eg on trajectory stitching), missing experiments on modern suites (eg ogbench), and complexity of the proposed method, with computational cost not fully analyzed. I think the authors thoroughly addressed all these concerns in the rebuttal. I am thus recommending the paper is accepted and encourage the authors to revise the manuscript following the reviewers' feedback and all new content provided in the rebuttal.